# Nanoscopic anatomy of dynamic multi-protein complexes at membranes resolved by graphene-induced energy transfer

Nadia Füllbrunn[1,2†], Zehao Li[1,2,3†], Lara Jorde[4], Christian P Richter[1], Rainer Kurre[1,2], Lars Langemeyer[1], Changyuan Yu[3], Carola Meyer[2,4], Jörg Enderlein[5,6], Christian Ungermann[1,2*], Jacob Piehler[1,2*], Changjiang You[1,2*]

[1]Department of Biology/Chemistry, University of Osnabrück, Osnabrück, Germany; [2]Center of Cellular Nanoanalytics (CellNanOs), University of Osnabrück, Osnabrück, Germany; [3]College of Life Sciences, Beijing University of Chemical Technology, Beijing, China; [4]Department of Physics, University of Osnabrück, Osnabrück, Germany; [5]3rd Institute of Physics - Biophysics, Georg August University, Göttingen, Germany; [6]Cluster of Excellence "Multiscale Bioimaging: from Molecular Machines to Networks of Excitable Cells" (MBExC), Georg August University, Göttingen, Germany

*For correspondence:
cu@uos.de (CU);
piehler@uos.de (JP);
cyou@uni-osnabrueck.de (CY)

†These authors contributed equally to this work

**Abstract** Insights into the conformational organization and dynamics of proteins complexes at membranes is essential for our mechanistic understanding of numerous key biological processes. Here, we introduce graphene-induced energy transfer (GIET) to probe axial orientation of arrested macromolecules at lipid monolayers. Based on a calibrated distance-dependent efficiency within a dynamic range of 25 nm, we analyzed the conformational organization of proteins and complexes involved in tethering and fusion at the lysosome-like yeast vacuole. We observed that the membrane-anchored Rab7-like GTPase Ypt7 shows conformational reorganization upon interactions with effector proteins. Ensemble and time-resolved single-molecule GIET experiments revealed that the HOPS tethering complex, when recruited via Ypt7 to membranes, is dynamically alternating between a 'closed' and an 'open' conformation, with the latter possibly interacting with incoming vesicles. Our work highlights GIET as a unique spectroscopic ruler to reveal the axial orientation and dynamics of macromolecular complexes at biological membranes with sub-nanometer resolution.

## Introduction

Numerous fundamental cellular processes such as energy conversion, signal transduction, and transport occur in the context of lipid membranes. In animals, approximately one-third of the proteome is localized at the various membranes of the cell, while more than 50% of the currently available pharmaceuticals target membrane proteins (*Santos et al., 2017*). The mechanistic understanding of these proteins has been mainly addressed by techniques that arrest them in particular conformations, and take them out of their biological membrane context (*Fernandez-Leiro and Scheres, 2016*; *Kosol et al., 2013*). Even more critical, we largely lack techniques to address the orientation and organization of large and highly flexible protein complexes at membranes involved in signaling or interorganellar communication and fusion.

The structurally highly heterogeneous and dynamic systems are extremely difficult to tackle by traditional structural techniques that average ensembles. Single-molecule Förster resonance energy

**eLife digest** Proteins are part of the building blocks of life and are essential for structure, function and regulation of every cell, tissue and organ of the body. Proteins adopt different conformations to work efficiently within the various environments of a cell. They can also switch between shapes.

One way to monitor how proteins change their shapes involves energy transfer. This approach can measure how close two proteins, or two parts of the same protein, are, by using dye labels that respond to each other when they are close together.

For example, in a method called FRET, one dye label absorbs light and transfers the energy to the other label, which emits it as a different color of light. However, FRET only works over short distances (less than 10nm apart or 1/100,000th of a millimeter), so it is not useful for larger proteins.

Here, Füllbrunn, Li et al. developed a method called GIET that uses graphene to analyze the dynamic structures of proteins on membrane surfaces. Graphene is a type of carbon nanomaterial that can absorb energy from dye labels and could provide a way to study protein interactions over longer distances.

Graphene was deposited on a glass surface where it was coated with single layer of membrane, which could then be used to capture specific proteins. The results showed that GIET worked over longer distances (up to 30 nm) than FRET and could be used to study proteins attached to the membrane around graphene. Füllbrunn, Li et al. used it to examine a specific complex of proteins called HOPS, which is linked to multiple diseases, including Ebola, measuring distances between the head or tail of HOPS and the membrane to understand protein shapes. This revealed that HOPS adopts an upright position on membranes and alternates between open and closed shapes.

The study of Füllbrunn, Li et al. highlights the ability of GIET to address unanswered questions about the function of protein complexes on membrane surfaces and sheds new light on the structural dynamics of HOPS in living cells. As it allows protein interactions to be studied over much greater distances, GIET could be a powerful new tool for cell biology research. Moreover, graphene is also useful in electron microscopy and both approaches combined could achieve a detailed structural picture of proteins in action.

transfer (smFRET) has been used successfully to shed light on the structural heterogeneity of proteins lacking structural definition (*Roy et al., 2008*; *Yang et al., 2018*; *Zhu et al., 2017*). However, with its limited dynamic range covering <10 nm as well as the need to introduce both donor and acceptor dyes with high fidelity and efficiency often obstructs its application. Membrane protein complexes, which often involve intrinsically disordered proteins or phase separation causing large-scale spatial re-arrangement, are therefore not amenable to smFRET. Single-molecule localization microscopy in turn is limited by axial localization precision of ~10 nm (*Gwosch et al., 2020*; *Schmidt et al., 2008*; *Shtengel et al., 2009*). Thus, single-molecule-based techniques providing sufficient spatiotemporal resolution in the 10–20 nm regime, which is relevant for multi-protein complexes, are currently not available.

Here, we introduce a novel approach for quantifying the axial organization and dynamics of large membrane protein complexes based on distance-dependent fluorescence quenching by graphene. This phenomenon is caused by radiation-less electromagnetic coupling of the excited fluorophore with graphene plasmons, which decays with the axial distance $d$ by $d^{-4}$ (*Swathi and Sebastian, 2009*). The atomic thickness of graphene ensures high optical transparency and an extremely high confinement of plasmons ($10^6$ times the diffraction limit) (*Koppens et al., 2011*). Compared to metal-induced energy transfer (MIET) (*Chizhik et al., 2014*), with a dynamic range that extends over a distance of ~150 nm, graphene-induced energy transfer (GIET) occurs within a dynamic range of ~30 nm, thus covering the dimensions of large membrane protein complexes (*Ghosh et al., 2019*). For exploiting GIET to probe axial organization of proteins in the context of membranes, we here developed a lipid monolayer assembly on graphene for site-specific protein capturing. We confirmed the theoretical distance-dependence of GIET within a dynamic range of 25 nm on graphene-supported lipid monolayers using DNA oligonucleotides as a nanoscale ruler. We successfully applied this approach for unraveling the axial organization and dynamics of large multi-protein

complexes involved in vesicular transport and fusion, which is critically required for delivery of proteins and lipids in organellar homeostasis (*Bröcker et al., 2010*; *Yu and Hughson, 2010*). We reconstituted the entire protein machinery required for tethering late endosomal vesicles, autophagosomes, and AP-3 vesicles to the vacuolar target membrane (*Bröcker et al., 2010*; *Nickerson et al., 2009*), including the Rab7-like Ypt7, its guanine nucleotide exchange factor (GEF) Mon1-Ccz1 and the homotypic fusion and vacuole protein sorting (HOPS) complex (*Brett et al., 2008*; *Nordmann et al., 2010*; *Ostrowicz et al., 2010*). In animal cells, HOPS is responsible for autophagy, the infectivity of Ebola virus, and linked to multiple diseases (*van der Beek et al., 2019*). Based on ensemble and single-molecule GIET, we quantitatively unraveled the axial organization and dynamics of Ypt7 and its interacting HOPS complex. Our data reveal that HOPS adopts an upright orientation on membranes with characteristic axial dynamics. We thus introduce GIET as a powerful novel technique to uncover the nanoscale spatiotemporal architecture of extended multiprotein complexes at membranes.

## Results

### Functional protein capturing onto graphene-supported lipid monolayers

To apply GIET to explore the structural and functional organization of protein complexes at membranes, we established lipid monolayer coating of graphene. Solid-supported graphene monolayers were prepared by transferring commercially available graphene sheets onto glass substrates. Coating of lipid on graphene was carried out by either liposome fusion or solution-assisted lipid deposition as reported previously (*Blaschke et al., 2018*; *Lima et al., 2016*; *Tabaei et al., 2016*). For site-specific capturing of His-tagged proteins, tris-nitrilotriacetic acid (trisNTA) moieties were incorporated into the lipid monolayer. For this purpose, vesicles made from 1,2-dioleoyl-sn-glycero-3-phosphocholine (DOPC) containing 5% trisNTA conjugated with dioctadecyl amine (trisNTA-DODA) (*Beutel et al., 2014*; *Lata et al., 2005*) were fused on freshly prepared graphene slides (*Figure 1A*). Lipid monolayer formation, protein immobilization, and interactions were monitored in real-time using simultaneous total internal reflection fluorescence spectroscopy and reflectance interference (TIRFS-RIF) detection (*Gavutis et al., 2005*). A mass signal of lipid deposition on graphene of 2.5 ng/mm$^2$ was observed after washing out excess vesicles (*Figure 1B*). For trisNTA-functionalized lipid bilayer formation by vesicle fusion on a silica surface, which has been previously established for protein interaction analysis at membranes (*Beutel et al., 2014*; *Gavutis et al., 2005*; *Lata et al., 2006*), a lipid deposition of 5 ng/mm$^2$ was observed (*Figure 1—figure supplement 1*). These results confirmed formation of a lipid monolayer on the hydrophobic graphene surface. Stable, Ni(II) ion specific immobilization of an anti-GFP nanobody fused to a C-terminal His-tag (NB-H6) was observed on the graphene-supported lipid monolayer. Upon injection of monomeric enhanced green fluorescence protein (mEGFP), we detected fast binding to the immobilized NB (*Figure 1B*). The same binding assays were carried out with silica-supported lipid bilayers, yielding very similar binding kinetics for NB-H6 and mEGFP (*Figure 1—figure supplement 1*). Furthermore, quantitative removal of immobilized protein by imidazole was observed (*Figure 1B*), which enabled for repeated immobilization. These experiments verified the functional integrity of His-tagged proteins captured onto graphene-supported lipid monolayers. The interaction with mEGFP was simultaneously quantified by TIRFS detection (*Figure 1C*). Comparing the mEGFP fluorescence signal observed on graphene ($I_G$) to that on the silica substrate ($I_0$) confirmed strong quenching by GIET with an efficiency of 83.6% for mEGFP bound to NB-H6. Direct immobilization of H6-tagged mEGFP on graphene resulted into even stronger GIET (*Figure 1D*) with an efficiency of 92.4%. As the size of NB is ~2 nm based on the crystal structure (*Kirchhofer et al., 2010*), these results highlight that GIET can reveal nanoscale distances of proteins immobilized on a graphene-supported lipid monolayer.

### DNA nanoruler calibration confirms distance-dependent GIET efficiency

To determine axial distances from the membrane, we calibrated the distance-dependent GIET efficiencies by using fluorescent-labeled DNA strands attached to the lipid monolayer. Four single strand DNA (ssDNA) with 20mer, 25mer, 35mer, and 50mer nucleotides were designed as the scaffold of nanorulers (*Figure 2A*, DNA sequences in *Supplementary file 1A*). These ssDNAs share a common 20mer sequence at the 5´-end and cholesterol modification at the 3´-end for anchoring into

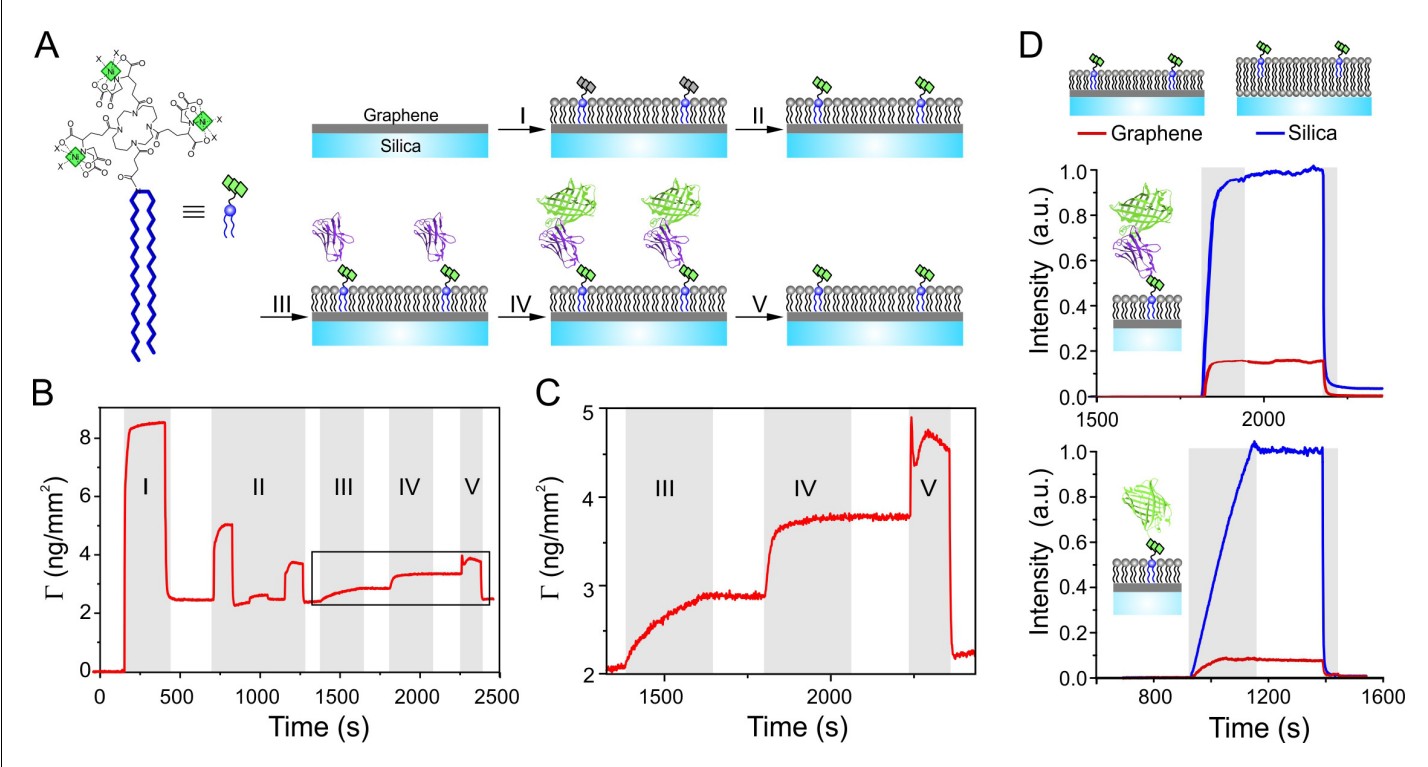

**Figure 1.** Functional protein reconstitution onto graphene-supported lipid monolayers. (**A**) Scheme of the graphene-supported lipid monolayer doped with trisNTA-DODA for site-specific capturing of His-tagged proteins. (**B, C**) Label-free detection of protein binding on graphene monitored by reflectance interference (RIF): (I) lipid coating; (II) conditioning of tris-NTA chelator by EDTA, Ni$^{2+}$, imidazole; (III) binding of anti-GFP nanobody fused with H6 tag (NB-H6); (IV) binding of tagless mEGFP; (**V**) imidazole wash. (**C**) Zoom-up showing immobilization of NB-H6 and interaction with tagless mEGFP. (**D**) Fluorescence quenching by GIET quantified in real-time by simultaneous TIRFS-RIF detection for mEGFP using trisNTA-functionalized lipid monolayer on graphene (red) or lipid bilayer on silica (blue), respectively. Upper panel: normalized fluorescence intensities for tagless mEGFP binding to immobilized NB-H6 (as in panels A–C). Lower panel: Fluorescence intensities of H6-mEGFP directly bound to trisNTA-DODA. Normalized fluorescence intensities were calculated according to *Equation (1)* taking the mass signals into account.

The online version of this article includes the following figure supplement(s) for figure 1:

**Figure supplement 1.** Surface sensitive TIRFS-RIF detection of protein binding on graphene.

the lipid monolayer (*Johnson-Buck et al., 2014*) (anchor strand). To promote, by electrostatic repulsion, perpendicular orientation of oligonucleotides with respect to lipid layer, 5% negatively charged 1,2-di-(9Z-octadecenoyl)-*sn*-glycero-3-phospho-*L*-serine (DOPS) was incorporated into the DOPC matrix. The probe strands have a common 20mer ssDNA sequence complementary to the anchor strand and were conjugated with 6-carboxyfluorescein (FAM) at either 3´- or 5´- end, respectively, for fluorescence read-out. By hybridization of the probe and anchor strands, eight distinct fluorophore distances from the monolayer surface were obtained, which are denoted as anchor strand-3´F or 5´F. For the 35mer and 50mer anchor strands, additional complementary 15mer and 30mer ssDNA were used as blockers, respectively, to obtain the fully length double strand DNA nanorulers (*Figure 2A*). For the 25mer anchor strand, a 5mer unpaired region remained after hybridization with the probe strands, which may enhance the flexibility of this system.

Using real-time, surface sensitive detection of TIRFS-RIF, we monitored formation of the lipid monolayer on graphene, integration of anchor strands, and hybridization with probe strands (*Figure 2—figure supplement 1*). For all probe-anchor hybridizations, detection of combination 20–5´F was not feasible, probably due to the steric hindrance upon inserting a FAM dye proximal to the lipid head groups. Instead, we used 2′,7′-difluorofluorescein-1,2-dihexadecanoyl-*sn*-glycero-3-phosphoethanolamine (OG488-DHPE), in which the dye molecule is directly linked to the lipid head group. In parallel, the same experiments were performed using silica-supported lipid bilayers. Simultaneous quantification of mass deposition and fluorescence intensity on both substrates ensured

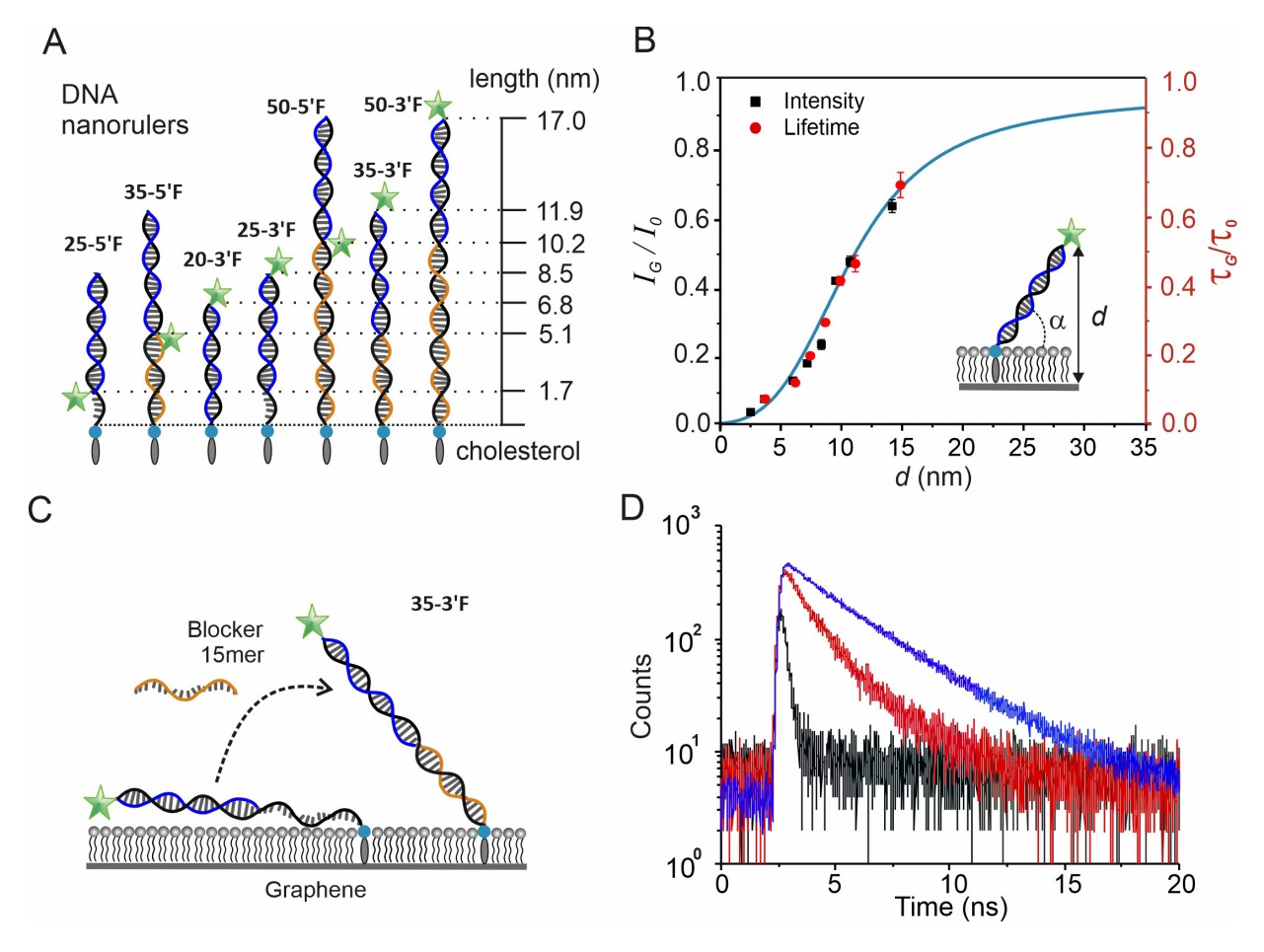

**Figure 2.** Calibration of distance-dependent GIET using DNA nanorulers. (**A**) Architecture and theoretical distances of membrane-anchored DNA nanorulers. Anchor strands (black) are modified with a 3′-end cholesterol. Probe strands (blue) are conjugated with fluorescein at either 3′- or 5′- end. Blocker strand (orange) for complementing the unpaired regions. Nomenclature is based on combinations of anchor and probe strands. (**B**) Calibration obtained from intensity ratios (black squares, black axis) and from lifetime ratios (red dots, red axis) as a function of distance from the fluorophore to graphene. Calibration at 2.5 nm by lifetime is not available due to detection limitation. The inset illustrates the model used for calculating the vertical distance $d$ with a globally fitted tilting angle $\alpha$ = (43 ± 1)°. Values are mean ±s.d. from three independent measurements. The simulated distance-dependent relation for FAM dye is shown in cyan. (**C**) Schematic illustration of the 35mer ssDNA anchor strand's conformational changes on graphene. (**D**) Fluorescence lifetime histogram of the probe strand bound to the 35mer anchor DNA strand in the absence (black) and presence (red) of the 15mer blocker strand. For comparison, the experiment on a glass-supported lipid bilayer is shown (blue).

The online version of this article includes the following figure supplement(s) for figure 2:

**Figure supplement 1.** Surface sensitive TIRFS-RIF detection for GIET calibration.

**Figure supplement 2.** Simulation of electrodynamic coupling of an excited fluorophore to graphene.

**Figure supplement 3.** Time-correlated single photon counting (TCSPC) for determining the fluorescence lifetime of DNA nanorulers.

**Figure supplement 4.** Mono-exponential fitting of the TCSPC data of DNA rulers.

**Figure supplement 5.** Characterization of membrane mobilities on glass and graphene by fluorescence recovery after photobleaching (FRAP).

reliable quantification of GIET efficiencies (*Supplementary file 1B*). Significantly reduced fluorescence intensities on graphene ($I_G$) were observed compared to those on silica ($I_0$). The $I_G/I_0$ ratios increased with the end-to-end distances of the seven different FAM positions on DNA nanorulers. However, the increase was significantly lower than the theoretical prediction from electromagnetic simulations, which assumed fully perpendicular orientation on top of the lipid monolayer coating (Materials and methods: Electromagnetic simulation of GIET, and *Figure 2—figure supplement 2* for sensitivity analysis). Since we have previously validated our simulations using well-defined silica adlayers on the graphene surface (*Ghosh et al., 2019*), we wondered whether tilting of DNA strands

on surface was the reason for the discrepancy (*Kabeláč et al., 2012*; *Wong and Pettitt, 2004*). By assuming a tilting angle α between DNA and graphene, we calculated the vertical distance *d* by taking the 2.5 nm thickness of the lipid monolayer into account (*Attwood et al., 2013*; *Blaschke et al., 2018*) (Materials and methods: Validation of distance-dependent GIET by DNA nanorulers). Strikingly, the obtained correlation of $I_G/I_0$ vs *d* matches the predicted GIET curve well with a globally fitted tilting angle of (43 ± 1)° (*Figure 2B*), confirming the distance-dependent GIET on graphene-supported lipid monolayer. We also observed systematic deviations to higher experimental intensities at very short distances including OG488-DHPE, which may be ascribed to the fluorophores being attached via flexible linkers take a more distant position from the surface due repulsion from the negatively charged membrane surface.

## Axial distances can be reliably quantified via fluorescence lifetimes

GIET efficiencies were furthermore quantified by time-correlated single photon counting (TCSPC) using confocal laser-scanning microscopy (cLSM, *Figure 2—figure supplement 3*). Lifetime histograms could be nicely fitted with mono-exponential decay functions in all cases (*Figure 2—figure supplement 4*), indicating a homogenous structural organization of the DNA rulers. On glass surfaces, constant fluorescence lifetimes of FAM were observed for DNA nanorulers, either 2.8 ns for 3′ labeling or 3.3 ns for 5′ labeling (*Figure 2—figure supplement 4*). The shorter lifetime at 3′-end could be attributed to quenching by the adjacent G-C base pair as reported previously (*Nazarenko et al., 2002*). Ratios of fluorescence lifetimes on graphene ($\tau_G$) to those on glass ($\tau_0$) closely matched the corresponding intensity ratios, which further corroborated the simulated distance-dependent GIET efficiency (*Figure 2B*, *Supplementary file 1C*). Results of both intensity and lifetime measurements showed a dynamic range within 3–25 nm corresponding to changes in GIET efficiency from 93–15%. A significant difference was detected for nanorulers between 35–5′F and 35–3′F, confirming that GIET is robust for detecting distance changes of ~1.2 nm between 6 nm and 11 nm.

These consistent calibrations via fluorescent intensities and lifetime measurements corroborated the validity of our GIET model, with systematic deviations likely to be related to positional uncertainty caused by inherent flexibility of the nanoruler system. We therefore applied the theoretical GIET curve for subsequent quantification of distances from experimental $I_G/I_0$. For the axial organization of mEGFP and mEGFP:NB complex tethered to the DOPC monolayer surface via trisNTA-DODA, distances 1.3 nm for H6-mEGFP directly tethered to the membrane ($I_G/I_0$: 7.6%) and 3.0 nm for mEGFP captured via NB-H6 ($I_G/I_0$: 16.4%) were estimated. The distance of 1.7 nm induced by the nanobody is in good agreement with the crystal structure of the GFP-NB complex (*Kirchhofer et al., 2010*), highlighting the nanometer sensitivity of quantifying the axial position of proteins onto lipid monolayers by GIET.

## Axial reorganization at the lipid monolayer surface can be quantified by GIET

We next asked whether GIET is capable to measure distance changes larger than 10 nm. For this purpose, we chose the 35mer anchor strand with a fully extended length of 11.9 nm from 3′-end to 5′-end (*Figure 2C*). Upon hybridizing with the 20mer 3′ FAM probe strand, a fluorescence lifetime $\tau_G$ of 0.24 ± 0.02 ns was detected (*Figure 2D*). Comparing to $\tau_0$ of 3.04 ns on glass, the obtained $\tau_G/\tau_0$ ratio of 8.0 ± 0.6% corresponds to a distance of 4.4 ± 0.2 nm from the graphene (1.9 nm from the lipid headgroups), suggesting that the hybridized probe-anchor DNA strand collapses onto the surface of the lipid monolayer due to the flexible, unpaired 15mer gap. To test the hypothesis, a 15mer ssDNA blocker strand was added, resulting into a ~ sixfold increase of the fluorescence lifetime (1.50 ± 0.03 ns, *Figure 2D*). The $\tau_G/\tau_0$ ratio of 49 ± 1.0% corresponds to a distance of 10.9 ± 0.2 nm, that is a height of 8.4 nm above the lipid layer. Compared to the length of the 35mer DNA nanoruler (11.9 nm), the height corresponds to a tilting angle α = 45°, which is in good agreement with the tilting angle α obtained for the calibration. This observation not only supports the validation of the calibration, but also suggests loss in flexibility upon hybridization with the 15mer blocker strand forces the full duplex strand into an upstanding position (*Figure 2C*). These results demonstrate the potential of GIET to quantify large-scale conformational rearrangements of biomolecules on membrane surfaces with nanometer precision. However, we did not observe lateral

diffusion dynamics of anchored DNA strands as explored by fluorescence recovery after photo-bleaching (FRAP, *Figure 2—figure supplement 5*), thus ensuring that changes in fluorescence life-time are not related to changes in the lateral organization within the lipid monolayer.

## Reconstitution of the HOPS tethering axis for structural analysis by GIET

We applied our new method to unravel the orientation and organization of proteins involved in vesi-cle tethering and fusion at the yeast vacuole. This process is initiated by the Rab7-like GTPase Ypt7 as a functional marker of the late endosomal membrane (*Figure 3A*). Upon activation by its guanine nucleotide exchange factor (GEF) Mon1-Ccz1 on endosomes and vacuoles (*Nordmann et al., 2010*), Ypt7-GTP recruits the 650 kDa heterohexameric HOPS tethering complex (*Brett et al., 2008*; *Ostrowicz et al., 2010*). This complex then tethers Ypt7-bearing membranes (*Hickey and Wickner, 2010*; *Ho and Stroupe, 2015*; *Lürick et al., 2017*; *Orr et al., 2015*), and catalyzes fusion of vacuoles or SNARE-carrying proteoliposomes (*Stroupe et al., 2009*; *Wickner and Haas, 2000*). Our structural studies revealed that HOPS adopts a flexible tadpole-like conformation (*Bröcker et al., 2012*; *Lürick et al., 2017*). HOPS has four central subunits Vps11, Vps18, Vps16, and Vps33 (*Rieder and Emr, 1997*), which are flanked by Vps39 and Vps41 as Rab-specific subunits at its tail and head, respectively (*Brett et al., 2008*; *Bröcker et al., 2012*; *Ostrowicz et al., 2010*). However, the overall architecture and thus function of HOPS is controversial as two different structures have been observed by negative-stain electron microscopy (*Bröcker et al., 2012*; *Chou et al., 2016*). In its compact form, HOPS is about 30 nm in length and 10 nm in width (*Bröcker et al., 2012*), whereas the more open conformation of a second study suggests an even longer particle (*Chou et al., 2016*). Importantly, both structures were obtained from purified complexes in solution, bearing the ques-tion, how the functional complex is organized on membranes. To tackle this question by GIET, we reconstituted the entire tethering machinery on graphene-supported lipid monolayers using recom-binantly expressed and purified components (*Figure 3B,C*). Like all Rabs, Ypt7 has an N-terminal GTPase domain, followed by a hypervariable domain (HVD), and a C-terminal prenyl group (*Goody et al., 2017*). The HVD varies in length between different Rab proteins and is critical for functionality (*Li et al., 2014*). Due to its composition, it has been assumed that the HVD adopts an extended conformation and would position the N-terminal GTPase domain of the Rab away from the membrane (*Burguete et al., 2008*). However, this assumption has not been tested, neither alone nor in the presence of GEFs or effectors.

## GIET reveals an extended HVD of Ypt7 on membranes

To analyze the conformational organization of Ypt7 on membranes, we produced this protein N-ter-minally fused to mNeonGreen (mNeon-Ypt7). The purified protein was prenylated in vitro (mNeon-pYpt7) in the presence of the chaperone GDI (mNeon-pYpt7:GDI), and the complex was used for reconstitution into lipid layers (*Langemeyer et al., 2018*; *Thomas and Fromme, 2016*; *Figure 3B*, *Figure 3—figure supplement 1*). In parallel, we generated a strain with mNeon-pYpt7 in vivo and observed full complementation (*Langemeyer et al., 2020*). Full-length complexes of Mon1-Ccz1 and HOPS were purified to homogeneity and monodispersity (*Figure 3C*), and we have previously shown that both function together with prenylated Ypt7 to promote fusion of proteoliposomes (*Langemeyer et al., 2018*). To test membrane binding, we incubated mNeon-pYpt7:GDI with lipo-somes by destabilizing the interaction with GDI, and observed that Ypt7 recruitment required the prenyl anchor and GTP to prevent GDI-mediated extraction (*Figure 3D*). We further analyzed the ability of activated pYpt7 to interact with HOPS, and thus incubated pYpt7-GTP carrying liposomes with purified HOPS (*Figure 3E*). In agreement with previous work (*Lürick et al., 2017*; *Orr et al., 2015*), HOPS triggered liposome clustering in a pYpt7 and concentration-dependent manner (*Figure 3—figure supplement 2*).

Efficient transfer of prenylated mNeon-pYpt7 to lipid monolayers (on graphene) and bilayers (on glass) and homogeneous distribution were confirmed by cLSM imaging (*Figure 3—figure supple-ment 3*). Binding of Mon1-Ccz1 to lipid mono- and bilayers (*Cabrera et al., 2014*; *Poteryaev et al., 2010*) was confirmed by label-free RIF detection (*Figure 3—figure supplement 4A*). Recruitment of HOPS to membrane-anchored mNeon-pYpt7-GTP resulted into significant changes of its mobility and the organization into dot-like structures (*Figure 3—figure supplement 5*, *Video 1*, *Video 2*).

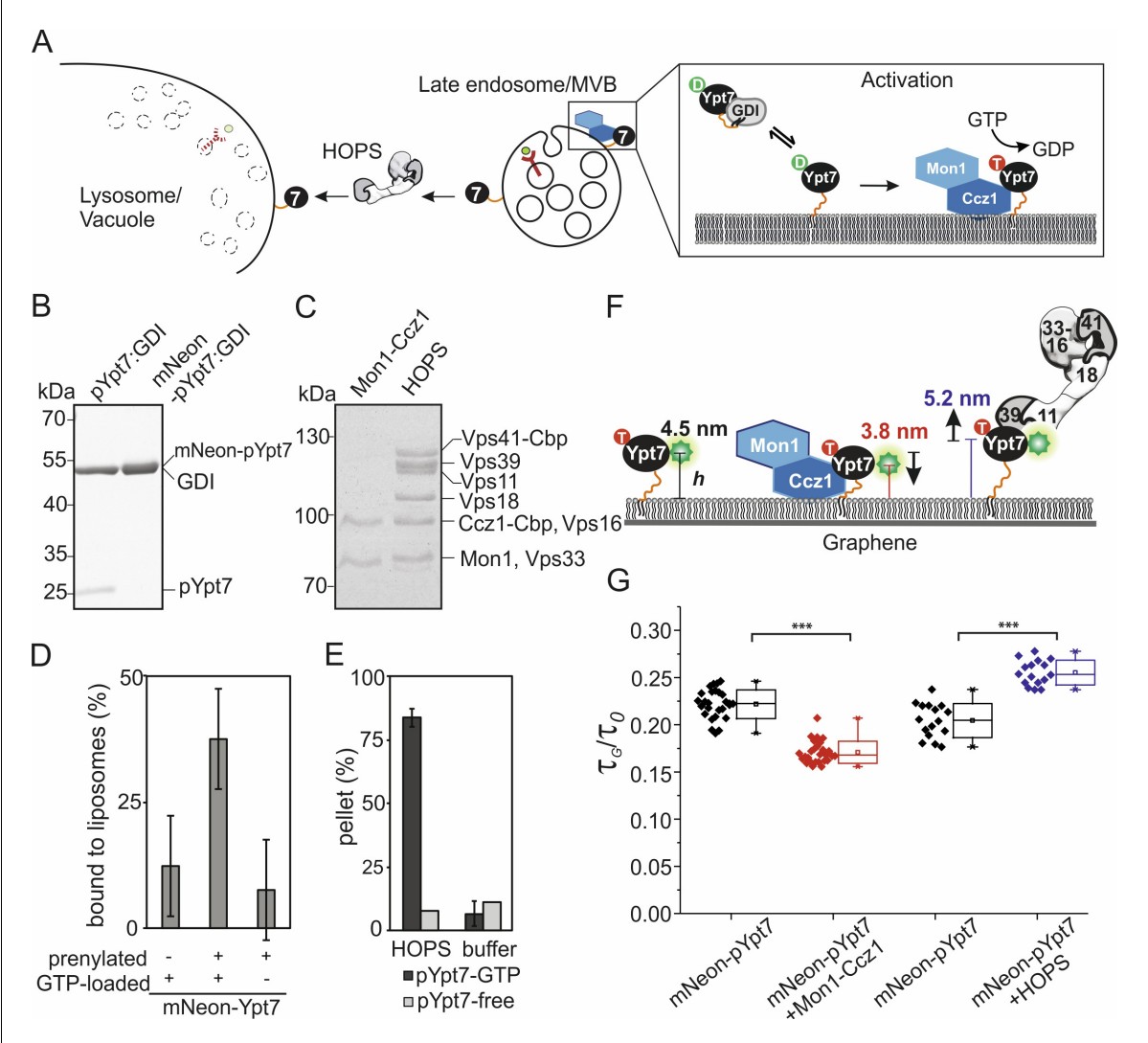

**Figure 3.** Conformational dynamics of membrane-associated mNeon-Ypt7 upon interaction with Mon1-Ccz1 and HOPS. (**A**) Schematic overview of delivery, activation, and function of Ypt7. GDI delivers prenylated GDP-bound Ypt7 to membranes of the late endosome/multivesicular body (MVB). Once Ypt7 is activated by its specific GEF Mon1-Ccz1 through GDP to GTP exchange, it stably associates with membranes. Activated Ypt7 recruits effector proteins, for example the HOPS tethering complex at the vacuolar membrane. (**B**) In vitro prenylation of Ypt7 (pYpt7) and mNeon-fused Ypt7 (mNeon-pYpt7) analyzed by SDS-PAGE and Coomassie staining (5 μg). (**C**) Tandem affinity purification of Mon1-Ccz1 and HOPS analyzed by SDS-PAGE and Coomassie staining (5 μg). (**D**) Membrane association of prenylated mNeon-Ypt7 with (middle bar) and without (left bar) GTP-loading. Fraction of membrane-bound mNeon-pYpt7 based on the fluorescent signal in the supernatant before and after sedimentation (n = 3). As a negative control, liposomes were incubated with unprenylated mNeon-Ypt7 in the presence of GTP (left bar). (**E**) Interaction of HOPS (350 nM) with pYpt7-GTP-loaded liposomes (dark gray bars) or pYpt7-free liposomes (light gray bars), respectively. The fraction of clustered liposomes in the pellet was calculated on the basis of fluorescent signal in the supernatant before and after sedimentation (n = 3). (**F**) Putative architecture of membrane-associated mNeon-pYpt7-GTP, and its complexes with Mon1-Ccz1 and HOPS, respectively. The flexible HVD domain is highlighted by an orange strand. (**G**) Box chart of fluorescence lifetime ratios of mNeon-pYpt7-GTP (black) on graphene-coated glass coverslip to glass ($\tau_G/\tau_0$), and its complexes with Mon1-Ccz1 (red) and HOPS (blue), respectively. Significance: ***$p<0.001$ (two t-test, n > 17).

The online version of this article includes the following figure supplement(s) for figure 3:

**Figure supplement 1.** Generation of mNeon-pYpt7-GDI complex.

**Figure supplement 2.** HOPS-mediated concentration-dependent tethering of liposomes loaded with His-fused Ypt7-GTP or prenylated Ypt7-GTP.

**Figure supplement 3.** Homogenous distribution of prenylated mNeon-Ypt7 in lipid mono- and bilayers.

**Figure supplement 4.** Characterization of Mon1-Ccz1 binding to membranes and fluorescence lifetime changes of mNeon-pYpt7.

**Figure supplement 5.** FRAP experiments of membrane-anchored mNeon-pYpt7 upon interaction with effectors.

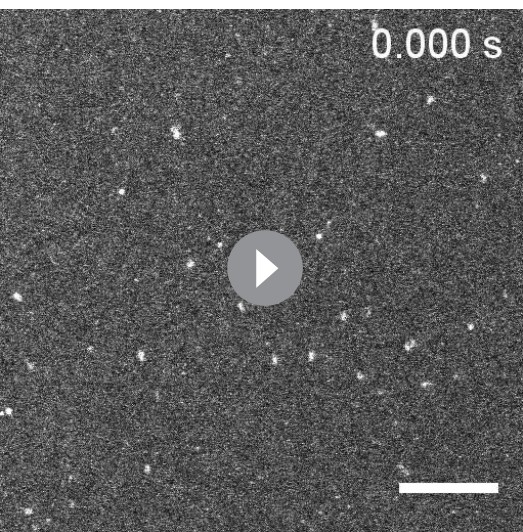

**Video 1.** Mobility of mNeon-Ypt7-GTP anchored into a glass-supported lipid bilayer probed by FRAP. Scale bar: 10 µm.

https://elifesciences.org/articles/62501#video1

Such lateral heterogeneity of mNeon-pYpt7-GTP: HOPS was not observed for graphene-supported monolayers, in line with the lack of lateral mobility of this architecture. By quantifying the fluorescence lifetime of mNeonGreen on graphene and glass (*Figure 3—figure supplement 4B*), the axial distances of active mNeon-pYpt7 from the membrane surface were determined based on the $\tau_G/\tau_0$ ratio in the absence of additional effector proteins and in the presence of Mon1-Ccz1 and HOPS (*Figure 3G*). pYpt7 was located surprisingly distal from the membrane in the absence of effector proteins with the mNeon fluorophore being located 4.5 ± 0.3 nm above the lipid monolayer surface. The site of mNeon fusion on Ypt7 is only ~1 nm away from the N-terminus of the HVD (*Wiegandt et al., 2015*) and therefore this observation suggests a rather extended conformation of the HVD. Upon interaction with Mon1-Ccz1 the distance of mNeon-pYpt7 decreased by 0.7 nm, while a further increase in the axial distance by 0.7 nm was observed in the presence of HOPS (*Figure 3F*). The total of 30% distance difference suggests the large conformational changes of the HVD are functionally relevant for the interaction of Ypt7 with each complex.

## The HOPS complex adopts an upright orientation upon binding Ypt7 on membranes

The HOPS complex binds Ypt7-GTP via its subunits Vps39 and Vps41 (*Brett et al., 2008*; *Ostrowicz et al., 2010*). Since these subunits have been mapped to opposite ends of the HOPS complex (*Bröcker et al., 2012*), HOPS may be oriented differently on membranes, depending whether it binds Ypt7 via one or both sites. We therefore systematically mapped the axial arrangement of the HOPS complex bound to membrane-inserted pYpt7-GTP by GIET. Thus, we used HOPS variants with C-terminal yeast-enhanced GFP (yEGFP) on Vps39, Vps11, Vps18, Vps16, and Vps33, respectively, and overexpressed and purified the complexes from yeast. Uncompromised complex assembly was verified by SDS-PAGE (*Figure 4A*). yEGFP-tagged HOPS was specifically captured to lipid mono- and bilayers only after anchoring pYpt7-GTP (*Figure 4B*). Significantly heterogeneous intensity distribution of yEGFP-tagged HOPS was observed on lipid bilayers but not on lipid monolayers on graphene (*Figure 4B*, *Figure 4—figure supplement 1*). These results suggest that HOPS clusters upon binding to diffusive membrane-anchored pYpt7-GTP, in line with the above-mentioned clustering and loss in mobility of mNeon-pYpt7-GTP upon binding to untagged HOPS (*Figure 3—figure supplement 5*).

To determine the axial orientation of Ypt7-bound HOPS on membranes, GIET efficiencies were quantified by measuring fluorescence lifetimes on graphene ($\tau_G$) and glass support ($\tau_0$).

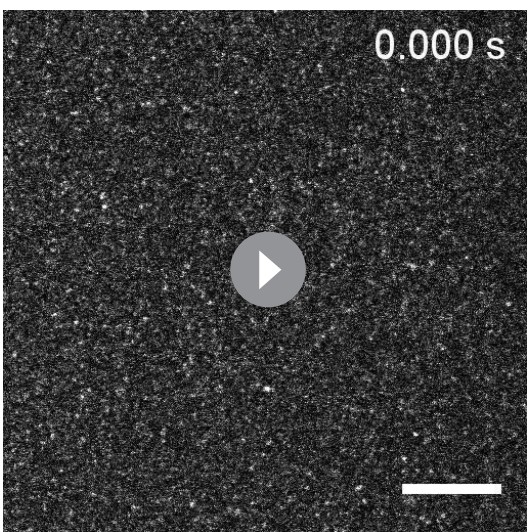

**Video 2.** FRAP of mNeon-Ypt7-GTP in interaction with HOPS on glass-supported lipid bilayer. Scale bar: 10 µm.

https://elifesciences.org/articles/62501#video2

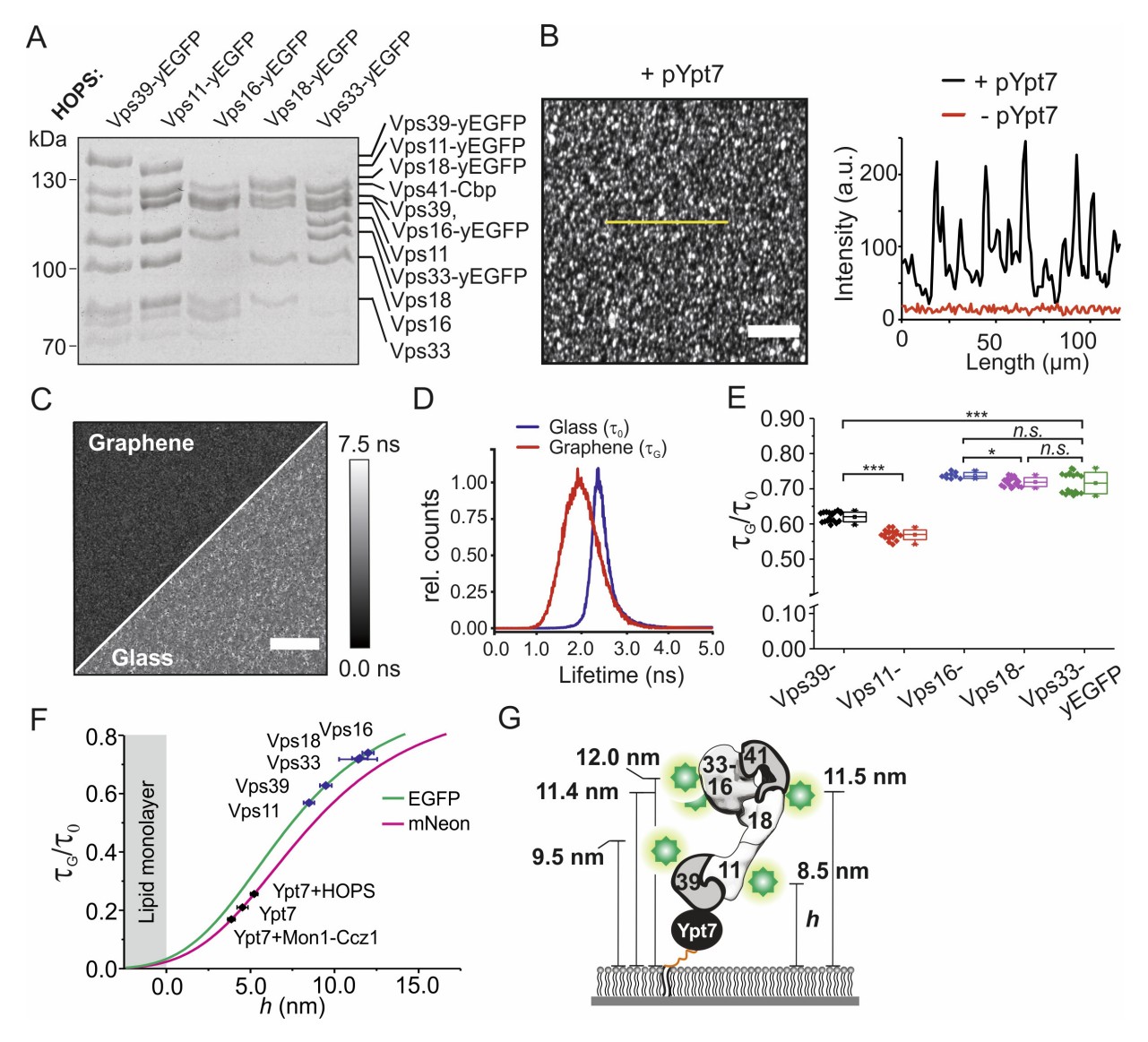

**Figure 4.** Axial architecture of pYpt7-bound HOPS on membranes explored by GIET. (**A**) Integrity of HOPS complexes containing different yEGFP-fused subunits analyzed by SDS-PAGE and Coomassie staining. (**B**) Confocal laser-scanning microscopy image of HOPS Vps16-yEGFP bound to pYpt7-GTP-loaded membrane on glass. Scale bar: 50 μm. Fluorescence intensity profile along the yellow line in the image is shown aside (black). As a negative control, fluorescence intensity profile of HOPS (Vps16-yEGFP) in the absence of pYpt7-GTP is shown (red). (**C**) Fluorescence lifetime imaging of HOPS (Vps16-yEGFP) bound to pYpt7-GTP-loaded membranes on glass and graphene, respectively. Scale bar: 50 μm. (**D**) Normalized fluorescence lifetime histograms of FLIM images shown in panel C. (**E**) Fluorescence lifetime ratios of different yEGFP-fused subunits in HOPS complex. The HOPS complexes were bound to pYpt7-GTP-loaded membranes on glass and graphene, respectively. Significance: ***p<0.001 (two t-test, n > 9), *p<0.05 (two t-test, n > 9). (**F**) Axial distances of yEGFP-fused HOPS subunits above membrane determined according to the GIET-distance correlation for EGFP (green). The range of distance is shown by mean ±s.d of the lifetime ratio. Axial distances of Ypt7 and Ypt7 bound to complexes are shown for reference (magenta, correlation curve of mNeon). (**G**) Putative orientation of pYpt7-GTP bound HOPS complex on membranes.

The online version of this article includes the following figure supplement(s) for figure 4:

**Figure supplement 1.** Fluorescence intensity imaging by cLSM and lifetime imaging by FLIM of the recruited HOPS complex.

**Figure supplement 2.** Fluorescence lifetimes of yEGFP-fused HOPS complex bound to membrane-anchored pYpt7-GTP on glass and graphene.

FLIM images of yEGFP-tagged HOPS bound to pYpt7-GTP showed homogeneous distribution on graphene, yielding a Gaussian-like distribution of fluorescence lifetimes (*Figure 4C,D*). Interestingly, $\tau_G$ showed a broader distribution than $\tau_0$, in line with some heterogeneity of the axial arrangement. The mean fluorescence lifetimes were used to estimate the average distance from the lipid head

**Table 1.** Conformational states characterized by HMM analysis of smGIET.

| Protein | HOPS Vps33-yEGFP | | | HOPS Vps11-yEGFP | | |
|---|---|---|---|---|---|---|
| State | L | M | H | L | M | H |
| $I_G/I_0$ [a] | 0.25 ± 0.06 | 0.41 ± 0.08 | 0.69 ± 0.13 | 0.22 ± 0.05 | 0.36 ± 0.06 | 0.51 ± 0.08 |
| $h$ (nm) [b] | 4.7 (3.9–5.5) | 6.8 (5.7–7.9) | 11.2 (8.9–14.9) | 4.3 (3.6–5.0) | 6.1 (5.3–6.9) | 8.1 (7.0–9.3) |
| Occup. (%) [c] | 30.9 ± 0.1 | 35.7 ± 0.2 | 33.5 ± 0.1 | 40.4 ± 0.2 | 38.0 ± 0.2 | 21.6 ± 0.1 |

[a]: mean ±s.d. based on $I_G$ of the Gaussian fits in single-molecule intensity distribution on graphene. $I_0$ is the mean value of Gaussian fit on glass. [b]: $h$ is the average height of NB-labeled HOPS on the membrane. Values in brackets are the range of $h$ determined by mean ±s.d of $I_G/I_0$. [c]: mean ± s.e.m. of state occupancy.

groups. For all yEGFP-fused subunits of HOPS, a significant decrease of $\tau_G$ compared to $\tau_0$ was found. The axial distances calculated for the different HOPS subunits are in line with an upright position of the complex bound to Ypt7 (*Figure 4E,F*, *Figure 4—figure supplement 2*): The two subunits at the putative HOPS tail, Vps39 (9.5 nm) and Vps11 (8.5 nm), were found to be much closer to the membrane than those of the previously annotated head (Vps16 at 12.0 nm, Vps18 at 11.4 nm, and Vps33 at 11.5 nm) (*Figure 4G*). These results suggest membrane anchoring of HOPS with membrane-anchored Ypt7-GTP occurs exclusively via interaction of Vps39, in line with its higher affinity as compared to Vps41 (*Lürick et al., 2017*; *Plemel et al., 2011*). The ~8 nm distance of the head subunits Vps33 and Vps16 from Ypt7-GTP found by GIET analyses, however, is significantly lower than the ~20 nm head-to-tail distance (between Vps33/16 and Vps39) observed in the EM structure of isolated HOPS (*Bröcker et al., 2012*). These findings suggest that the HOPS complex adopts a more compact structure upon docking to membrane-anchored Ypt7-GTP.

## Structural dynamics of HOPS complex at the membrane

Negative-stain EM studies revealed that the HOPS complex can adopt different conformations in solution (*Bröcker et al., 2012*; *Chou et al., 2016*), which is in line with the broad distribution of GIET efficiencies observed in fluorescence lifetime histograms (*Figure 4D*). To resolve such potential conformational heterogeneity of HOPS when bound to Ypt7 on membranes and possible transitions between these conformations, we turned to single-molecule imaging. The 97.7% transparency of graphene monolayer for visible light (*Nair et al., 2008*) allows single-molecule imaging on graphene by total internal reflection fluorescence (TIRF) microscopy with minimum loss of photons. For robust single-molecule GIET (smGIET) analysis, we used an anti-GFP nanobody site-specifically coupled to a photostable fluorophore ([Dy647]NB) at very low concentration (50 pM) to sub-stochiometrically label yEGFP-tagged subunits in the HOPS complex at the head (Vps33) and the tail (Vps11), respectively. Under these conditions, signals from specifically labeled individual HOPS complexes were detected by TIRF microscopy, as confirmed by photobleaching analyses on glass and graphene (*Figure 5A,B*, *Figure 5—figure supplement 1A*). Analysis of ~100 single step bleaching fluorescence traces yielded $I_G/I_0$ of 0.76 ± 0.20 for HOPS Vps33 and 0.53 ± 0.14 for HOPS Vps11 (*Figure 5—figure supplement 1B,C*). These mean values were in a good agreement with the ensemble lifetime measurements (*Supplementary file 2*). However, the large standard deviations corroborated significant structural heterogeneity in HOPS complexes that was already suggested by the broad fluorescence lifetime distribution observed in the ensemble experiments (*Figure 4D*).

Strikingly, single-molecule fluorescence of the HOPS complex bound to graphene-supported lipid monolayers showed pronounced fluctuations, which was not observed on glass

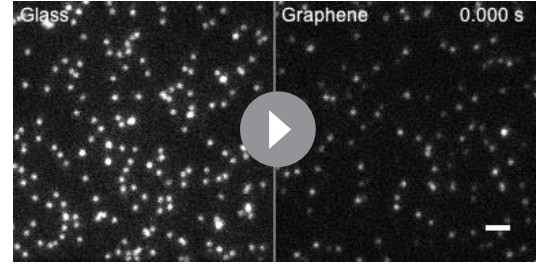

**Video 3.** Single-molecule fluorescence images of NB labeled HOPS Vps33-yEGFP on glass and graphene, respectively, imaged under the same conditions. Scale bar: 2 μm.
https://elifesciences.org/articles/62501#video3

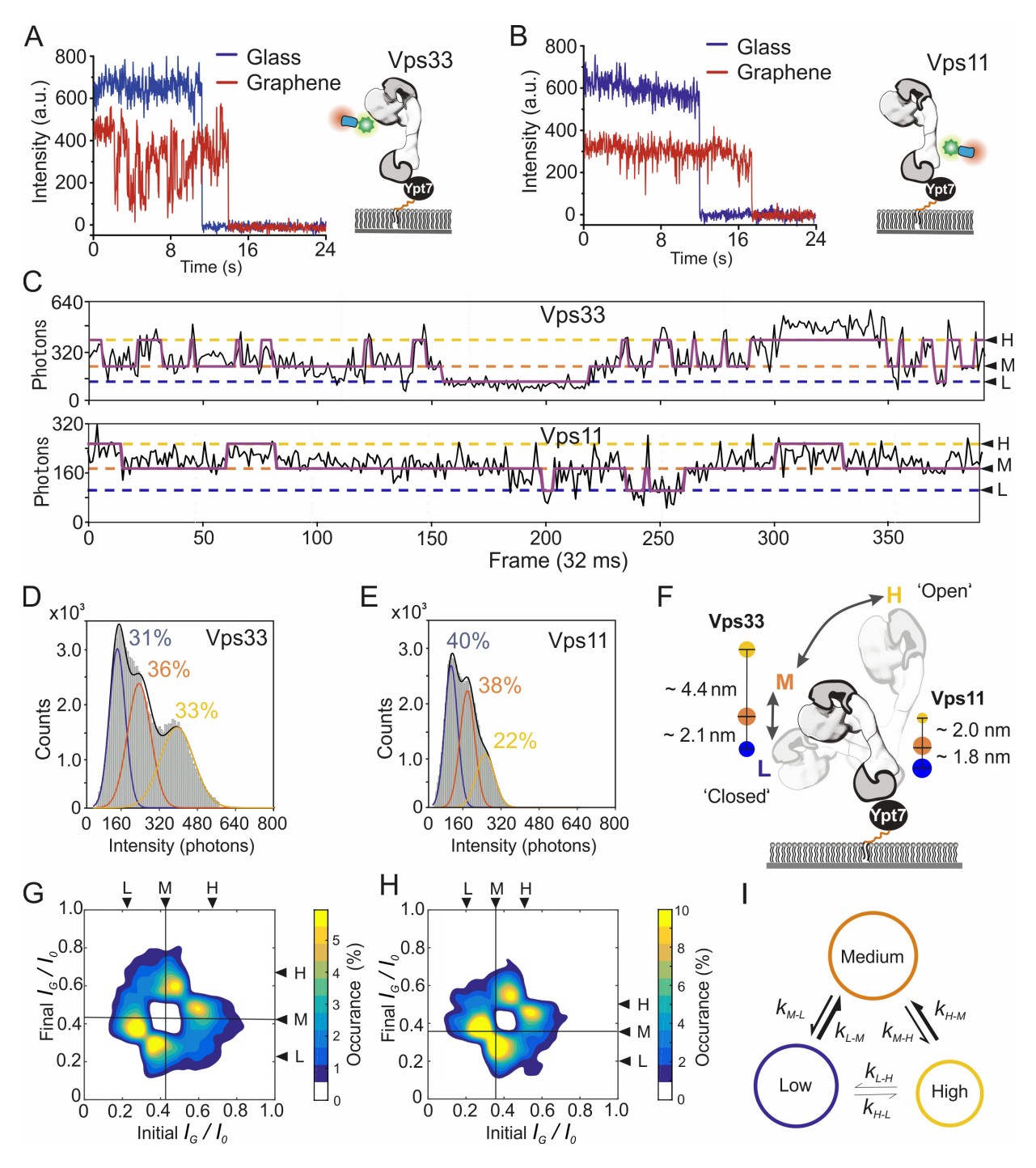

**Figure 5.** Dynamics of the pYpt7-bound HOPS complex on a lipid layer explored by single-molecule GIET. (**A, B**) Representative time-lapse single-molecule intensity traces of $^{Dy647}$NB-labeled HOPS Vps33-yEGFP (**A**) and HOPS Vps11-yEGFP (**B**) on glass (blue) and graphene (red). (**C**) Enlarged representation of the smGIET intensity fluctuations and fit by an HMM using a three-state segmentation. (**D, E**) Pooled single-molecule intensity distribution of $^{Dy647}$NB-labeled HOPS Vps33-yEGFP (D, n = 101615) and HOPS Vps33-yEGFP (E, n = 84218) immobilized on graphene-supported monolayers via Ypt7-GTP. Intensities were classified by Gaussian fits as states of low (L, blue), medium (M, orange), and high (H, yellow), respectively. (**F**) Model schematically depicting possible conformational changes of HOPS in the L, M, and H state. State occupancies and distance changes of Vps11 and Vps33 are indicated. (**G, H**) Transition density plots of HOPS Vps33-yEGFP (**G**) and Vps11-yEGFP (**H**). The crosslines on M state are guide for the eye. (**I**) Conformational transition kinetics obtained from HMM. Diameters of the circles and sizes of the arrows are proportional to the averaged state occupancy and transition rates from results of HOPS Vps33 and Vps11, respectively.

*Figure 5 continued on next page*

*Figure 5 continued*

The online version of this article includes the following figure supplement(s) for figure 5:

**Figure supplement 1.** Single-molecule detection of NB-labeled HOPS complexes bound to lipid-anchored pYpt7-GTP.

**Figure supplement 2.** Pooled single-molecule intensity histograms for (**A**) HOPS Vps33-yEGFP (n = 60498) and (**B**) HOPS Vps11-yEGFP (n = 33317) on glass, respectively.

**Figure supplement 3.** Representative root mean square deviation (RMSD) of single-molecule intensity traces on glass.

(*Figure 5A,B*, *Video 3*). These fluctuations therefore indicate axial conformational dynamics of pYpt7-GTP bound HOPS on the membrane, which is monitored faithfully by distance sensitive smGIET. To quantify the conformational dynamics of the HOPS complex, we recorded time-lapse intensity traces of individual HOPS complexes on graphene and glass (*Figure 5C*). HOPS Vps33-yEGFP was chosen based on its pronounced two lifetime populations observed on graphene in the ensemble experiments (*Figure 4E*, *Figure 4—figure supplement 2*). For pooled single-molecule intensities from traces of HOPS Vps33-yEGFP and HOPS Vps11-yEGFP, broad distributions were observed on graphene (*Figure 5D,E*) but not on glass (*Figure 5—figure supplement 2*). Analysis by Hidden Markov Modeling (HMM) (*McKinney et al., 2006*) identified three distinct states with 'low' (L), 'medium' (M), and 'high' (H) intensity, respectively, for both HOPS variants on graphene. The characteristic $I_G/I_0$ ratios, axial distances, and occupancies for each state are summarized in *Table 1*. The similar changes of mean axial distance from L to M state (~2 nm) in both HOPS Vps33-yEGFP and HOPS Vps11-yEGFP suggest a largely concerted axial movement of the entire HOPS complex. By contrast, the high discrepancy of distance changes during transitions from M to H (Vps33: 4.4 nm; Vps11: 2 nm) indicate a conformational re-organization of the HOPS complex itself, thus adopting an elongated conformation (*Figure 5F*). In total, the conformational dynamic distance range on the membrane was ~7 nm for HOPS Vps33-yEGFP and 4 nm for HOPS Vps11-yEGFP, respectively. Furthermore, we obtained rate constants of the transition between L, M, and H states from HMM analysis for HOPS Vps33-yEGFP and HOPS Vps11-yEGFP (*Supplementary file 4*). Striking similarity of the transition probability densities were obtained for both labeled subunits, with the dominant transitions between L-M and M-H states (*Figure 5G,H*). This observation supports correlated conformational dynamics of the L, M, and H states observed for HOPS Vps33-yEGFP and HOPS Vps11-yEGFP. Together, the time-resolved smGIET experiments reveal a 2-step conformational transition of the Ypt7-anchored HOPS complex from the 'closed' state L to the 'open' state H via an intermediate state M (*Figure 5I*).

## Discussion

Quantitative insights into the structural organization and dynamics of protein complexes and large machineries at membranes is important to understand their function. Methods based on fluorescence-interference-contrast (FLIC) microscopy (*Braun and Fromherz, 1998*) have been successfully employed to determine the axial organization of membrane proteins at ensemble level with at ~1 nm resolution (*Kiessling and Tamm, 2003*; *Kiessling et al., 2018*). Resolving the axial conformational dynamics that involved in many membrane-associated processes, however, requires methods that provide sub-nanometer resolution with single-molecule sensitivity at sub-second timescale. To tackle this challenge, we have here introduced fluorescence quenching by GIET using graphene-supported lipid monolayer as a membrane model. Detailed calibration with DNA rulers confirmed excellent correlation of simulated and measured distance-GIET relationships between 3 nm and 15 nm, promising very good sensitivity up to a distance of ~25 nm from the graphene surface, thus substantially exceeding the ~10 nm limit of FRET. Next to rapid and robust ensemble measurements, we demonstrate highly sensitive single-molecule GIET detection yielding relative axial localization precision of 10% at video rate temporal resolution (Materials and methods: Axial localization precision of smGIET). Thus, the spatial regime covered by GIET ideally fills the gap left by FRET and localization microscopy.

Proof-of-concept experiments demonstrate the feasibility to reconstitute complex membrane-anchored multi-protein machineries onto graphene-supported lipid monolayers and to quantify their axial architecture and dynamics. We show that the HVD domain of Ypt7 adopts a strongly extended

conformation, locating Ypt7 at a 4.5 nm distance above the membrane surface. This surprising observation could be explained by electrostatic repulsion of the negatively charged HVD. Significant changes in the axial positioning of Ypt7 were observed upon interaction with Mon1-Ccz1 and HOPS. Since Mon1-Ccz1 directly binds to the membrane (*Cabrera et al., 2014*), its Ypt7 binding site appears to be located distally to the membrane and the HVD allows unhindered access. Importantly, ensemble and single-molecule GIET furthermore revealed the nanoscale organization and dynamics of the HOPS tethering complex upon recruitment to the membrane surface. While ensemble GIET measurements clearly identified the HOPS subunit Vps39 as the primary binding site to lipid-anchored Ypt7, single-molecule GIET revealed rapid transition between three different conformational states with kinetics in the second to sub-second regime. Quantitative analysis of the kinetics suggests a 2-step transition from a 'closed' into an 'open' state involving concerted axial movement and conformational changes. Given the impaired diffusion in the graphene-supported lipid monolayer, however, the conformationally highly dynamic HOPS complex observed in our studies may represent an intermediate state prior to lateral HOPS clustering, which in turn could increase the efficiency of HOPS to tether incoming vesicles. By developing freely diffusive lipid architectures for graphene support, GIET will allow to dissect conformational organization and rearrangement of large protein complexes on membranes. Since GIET has negligible variation between spectrally different fluorophores (*Figure 2—figure supplement 2*), concerted conformational changes and axial movement can be potentially resolved by multicolor GIET. With graphene emerging as a routine support for transmission electron microscopy, correlative approaches using the same surface architecture can be envisioned.

# Materials and methods

## Key resources table

| Reagent type (species) or resource | Designation | Source or reference | Identifiers | Additional information |
|---|---|---|---|---|
| Gene (*Saccharomyces cerevisiae*) | mNeon-Ypt7 | Thermo Fisher Scientific | | |
| Strain, strain background (*S. cerevisiae*) | BY4732 | Euroscarf library | | MATa *his3Δ200 leu2Δ0 met15Δ0 trp1Δ63 ura3Δ0* |
| Strain, strain background (*S. cerevisiae*) | BY4727 | Euroscarf library | | MATalpha *his3Δ200 leu2Δ0 lys2Δ0 met15Δ0 trp1Δ63 ura3Δ0* |
| Strain, strain background (*S. cerevisiae*) | CUY2470 | doi: 10.1016/j. cub.2010.08.002 | | *BY4732; CCZ1::TRP1-GAL1pr MON1::HIS3M × 6-GAL1pr CCZ1::TAP-URA3* |
| Strain, strain background (*S. cerevisiae*) | CUY2675 | doi:10.1111/j. 1600-0854.2010. 01097.x | | *BY4732xBY4727 VPS41::TRP1-GAL1pr VPS41::TAP-URA3 VPS39::KanMX-GAL1pr VPS33::HIS3-GAL1pr VPS11::HIS3-GAL1pr VPS16::natNT2-GAL1pr VPS18::kanMX-GAL1pr-3HA* |
| Strain, strain background (*S. cerevisiae*) | CUY4391 | doi: 10.1073/ pnas. 1117797109 | | *BY4732xBY4727 VPS41::TRP1-GAL1pr VPS41::TAP-URA3 VPS39::KanMX-GAL1pr VPS39::yEGFP-hphNT1 VPS33::HIS3-GAL1pr VPS11::HIS3-GAL1pr VPS16::natNT2-GAL1pr VPS18::kanMX-GAL1pr-3HA* |
| Strain, strain background (*S. cerevisiae*) | CUY4392 | doi: 10.1073/ pnas. 1117797109 | | BY4732xBY4727 VPS41::TRP1-GAL1pr VPS41::TAP-URA3 VPS39::KanMX-GAL1pr VPS33::HIS3-GAL1pr VPS11::HIS3-GAL1pr VPS11::yEGFP-hphNT1 VPS16::natNT2-GAL1pr VPS18::kanMX-GAL1pr-3HA |

*Continued on next page*

*Continued*

| Reagent type (species) or resource | Designation | Source or reference | Identifiers | Additional information |
|---|---|---|---|---|
| Strain, strain background (*S. cerevisiae*) | CUY4393 | doi: 10.1073/pnas.1117797109 | | BY4732xBY4727 VPS41::TRP1-GAL1pr VPS41::TAP-URA3 VPS39::KanMX-GAL1pr VPS33::HIS3-GAL1pr VPS11::HIS3-GAL1pr VPS16::natNT2-GAL1pr VPS16::yEGFP-hphNT1 VPS18::kanMX-GAL1pr-3HA |
| Strain, strain background (*S. cerevisiae*) | CUY4394 | doi: 10.1073/pnas.1117797109 | | *BY4732xBY4727 VPS41::TRP1-GAL1pr VPS41::TAP-URA3 VPS39::KanMX-GAL1pr VPS33::HIS3-GAL1pr VPS11::HIS3-GAL1pr VPS16::natNT2-GAL1pr VPS18::kanMX-GAL1pr-3HA VPS18::yEGFP-hphNT1* |
| Strain, strain background (*S. cerevisiae*) | CUY4395 | doi: 10.1073/pnas.1117797109 | | *BY4732xBY4727 VPS41::TRP1-GAL1pr VPS41::TAP-URA3 VPS39::KanMX-GAL1pr VPS33::HIS3-GAL1pr VPS33::yEGFP-hphNT1 VPS11::HIS3-GAL1pr VPS16::natNT2-GAL1pr VPS18::kanMX-GAL1pr-3HA* |
| Recombinant DNA reagent | pET21a-EGFP | Novagen | | |
| Recombinant DNA reagent | pET21a-NB-H6 | Novagen | | |
| Recombinant DNA reagent | pET24b-Ypt7 | doi: 10.1242/jcs.140921 | | |
| Recombinant DNA reagent | pET24d-GST-TEV-Ypt7 | doi: 10.1091/mbc.e11-12-1030 | | |
| Recombinant DNA reagent | pET24d-GST-TEV-mNeon-Ypt7 | this paper | | mNeon-Ypt7 gene was synthesized by Thermo Fisher Scientific, provided in a pMA-T backbone and subcloned into a pET24d vector. |
| Recombinant DNA reagent | pGEX-6P-Gdi1 | doi: 10.1083/jcb.201608123 | | |
| Recombinant DNA reagent | pCDF-DUET-1-Bet2-Bet4 | doi: 10.1083/jcb.201608123 | | |
| Recombinant DNA reagent | pET30-Mrs6 | other | | Gift from K. Alexandrov laboratory, Institute for Molecular Bioscience, The University of Queensland, Australia |
| Sequenced-based reagent | 20mer anchor DNA oligonucleotide | IDT | | 5′- GATGAATGGTGGGTGAGAGG-3′ -TEG-Cholesterol |
| Sequenced-based reagent | 25mer anchor DNA oligonucleotide | IDT | | 5′- GATGAATGGTGGGTGAGAGGTGAGG-3′ -TEG-Cholesterol |
| Sequenced-based reagent | 35mer anchor DNA oligonucleotide | IDT | | 5′- GATGAATGGTGGGTGAGAGGTGA GGAGTAAGAGGA-3′-TEG-Cholesterol |
| Sequenced-based reagent | 50mer anchor DNA oligonucleotide | IDT | | 5′- GATGAATGGTGGGTGAGA GGTGAGGAGTAAGA GGATGTGTTAGAGGGATG-3′-TEG-Cholesterol |
| Sequenced-based reagent | 3′-FAM probe DNA oligonucleotide | IDT | | 5′-CCTCTCACCCACCATTCATC-3′-FAM |
| Sequenced-based reagent | 5′-FAM probe DNA oligonucleotide | IDT | | 5′- FAM-CCTCTCACCCACCATTCATC-3′ |
| Sequenced-based reagent | 15mer blocker DNA oligonucleotide | IDT | | 5′-TCCTCTTACTCCTCA-3′ |

*Continued on next page*

*Continued*

| Reagent type (species) or resource | Designation | Source or reference | Identifiers | Additional information |
|---|---|---|---|---|
| Sequenced-based reagent | 30mer blocker DNA oligonucleotide | IDT | | 5'-CATCCCTCTAACACATCCTC TTACTCCTCA-3′ |
| Peptide, recombinant protein | H6-mEGFP | doi: 10.1021/acs. nanolett. 5b01153 | | purified from *E. coli* BL21- DE3 cells |
| Peptide, recombinant protein | GPF NB 'enhancer' | doi: 10.1002/ smll.201502132 | | purified from *E. coli* BL21- DE3 cells |
| Software, algorithm | Origin8 | OriginLab | | |
| Software, algorithm | ImageJ | NIH | 1.53e | Time Series Analyzer plugin for extracting single-molecule intensity traces, Author: Balaji J http://rsb.info.nih.gov/ij/ plugins/time-series.html |
| Software, algorithm | MATLAB | Mathworks | R2019b | Code availability for calculating the GIET efficiency was documented in Nature Photonics 2019, 13: 860–865. doi: 10.1038/s41566-019-0510-7. |
| Software, algorithm | HMM | other | | Hidden Markov Model (HMM) Toolbox for Matlab written by Kevin Murphy https://www.cs.ubc.ca/~murphyk /Software/HMM/hmm.html |
| Software, algorithm | STaSI | doi: 10.1021/ jz501435p | | Algorithm of step transition and state identification for single-molecule data analysis. |

## Materials

Graphene monolayer was purchased from Graphenea, Spain (Easy Transfer Monolayer, G/P-25–25). Glass microscopy coverslips with thickness #1.5 and diameter of 24 mm were purchased from Carl Roth (PK26.1). 1,2-dioleoyl-sn-glycero-3-phosphocholine (DOPC), 1,2-dioleoyl-sn-glycero-3-phospho-serine (DOPS), 1-palmitoyl-2-oleoyl-sn-glycero-3-phosphocholine (POPC), 1-palmitoyl-2-oleoyl-sn-glycero-3-phosphatidylethanolamine (POPE), diacylglycerol (DAG) and DOGS-NTA were purchased from Avanti Polar Lipids, Alabama, USA. Phosphatidylinositol 3-phosphate (PI-3-P) was purchased from Echelon Biosciences Inc, Utah, USA. 2′,7′-difluoro-fluorescein conjugated with 1,2-dihexadeca-noyl-sn-glycero-3-phosphoethanolamine (OG488-DHPE) was purchased from Thermo Fisher Scientific. ATTO488-1,2-dipalmitoyl-sn-glycero-3-phosphoethanolamine (ATTO488-DPPE) was obtained from ATTO-TEC GmbH, Siegen, Germany. DY647P1 maleimide was purchased from Dyomics GmbH, Jena, Germany. TrisNTA-DODA was synthesized as previously reported (*Beutel et al., 2014*). DNA oligonucleotides were ordered from Integrated DNA Technologies, Inc The sequences and modifications are listed in *Supplementary file 1A*. Acetone was purchased from Merck (Uvasol for spectroscopy, 100022). Other reagents were purchased from Sigma Aldrich.

Protein expression, purification and labeling mEGFP with an N-terminal hexahistidine tag (H6-mEGFP) was cloned in the plasmid pET21a. The protein was expressed in *Escherichia coli* BL21 (DE 3) Rosetta cells, followed by purification of immobilized metal affinity chromatography (IMAC) and size exclusion chromatography (SEC) (*Wedeking et al., 2015b*). mEGFP without oligohistidine tag (tagless mEGFP) was expressed in *E. coli*. Lysed cells were heated to 80℃, centrifuged and the supernatant was purified by anion exchange column and SEC. Anti-GPF nanobody 'enhancer' (*Kirchhofer et al., 2010*) with a C-terminal hexahistidine tag (NB-H6) was expressed in *E. coli* Rosetta as described previously (*Wedeking et al., 2015a*). For fluorescence labeling, a C-terminal cysteine was appended to the NB for conjugating with Dy647P1-malemide. The anti-GFP nanobody 'enhancer' fused to a C-terminal hexahistidine tag was cloned into pET21a (pET21a-NB-H6) and

expressed in *E. coli* BL21 (DE 3) Rosetta cells. After cell lyses by sonication, NB-H6 was purified by immobilized metal affinity chromatography (IMAC) and size exclusion chromatography (SEC). For fluorescence labeling, a short linker including an additional cysteine residue followed by an ybbR-tag, a PAS repeat sequence and a terminal hexahistidine tag (a.a.: GSCGSGSKLDSLEFIASKLAPASPA SPASPASPASLEHHHHHH) was C-terminally fused to the NB. Expression and purification was performed as described for NB-H6. The purified proteins were reacted with a twofold molar ratio of DY647P1 maleimide for 30 min and then purified by size exclusion chromatography. A degree of labeling close to 1.0 was obtained as quantified by UV-Vis spectroscopy.

Endocytic proteins and protein complexes: (1) Rab GTPases and Gdi1. pET24b-Ypt7 (*Cabrera et al., 2014*), pET24d-GST-TEV-Ypt7 (*Lachmann et al., 2012*), pET24d-GST-TEV-mNeon-Ypt7 and pGEX-6P-Gdi1 (*Thomas and Fromme, 2016*) were transformed into competent *E. coli* BL21 (DE 3) Rosetta cells. Rab GTPases and Gdi1 were purified as previously described with slight modifications (*Langemeyer et al., 2018*; *Lürick et al., 2017*; *Nordmann et al., 2010*). The pET24b-Ypt7, pET24d-GST-TEV-Ypt7, pET24d-GST-TEV-mNeon-Ypt7 (mNeon fused to Ypt7 via a GGSGx3 linker) and pGEX-6P-Gdi1 were expressed in *E. coli* BL21 (DE 3) Rosetta cells. Cells were grown in Luria broth (LB) medium containing the specific antibiotics until an $OD_{600}$ of around 0.8, before they were induced with 0.25 mM isopropyl-β-D-thiogalactoside (IPTG) for 18 hr at 16°C. For purification of the Rab GTPases, harvested cells were lysed by a Microfluidizer, Model M-110L (Microfluidics, Newton, MA) in 50 mM HEPES, pH 7.5, 150 mM NaCl, 1 mM $MgCl_2$, 1 mM DTT, 1 mM phenylmethylsulfonyl fluoride (PMSF), and 0.05-fold protease inhibitor cocktail. For purification of Gdi1, lysis was performed in PBS supplemented with 5 mM β-mercaptoethanol and 1 mM PMSF. The lysate was centrifuged at 40,000 g for 30 min at 4°C, and the supernatant was incubated for 2 hr at 4°C with pre-equilibrated Ni-NTA agarose (Macherey-Nagel, Germany) for His-fused proteins or Glutathione Sepharose (GSH) 4B beads (GE Healthcare) for GST-fused proteins. Ni-NTA beads were washed with 50 ml lysis buffer lacking PMSF and protease inhibitor but supplemented with 10 mM imidazole. For elution, the imidazole concentration was increased to 300 mM. After elution, the buffer was exchanged via a PD10 column (GE Healthcare) containing no imidazole. GSH beads were extensively washed with 120 ml lysis buffer lacking PMSF and protease inhibitor cocktail, and proteins were eluted by cleaving the affinity-tag with TEV- or Precision-protease, respectively, for 2 hr at 16°C.

(2)Rab GGTase and Rab Escort Protein. pET30-Mrs6 (*Pylypenko et al., 2003*) and pCDF-Duet-1-Bet2-Bet4 (*Thomas and Fromme, 2016*) were transformed into *E. coli* BL21 (DE 3) Rosetta cells. The Rab GGTase and the Rab escort protein were expressed and induced as Rab GTPases and Gdi1. The Rab GGTase (pCDF-Duet-1-Bet2-Bet4) and the Rab escort protein (pET-Mrs6) were expressed and induced as described in 1.2. Harvested cells were lysed in 50 mM Tris, pH 8.0, 300 mM NaCl, 2 mM β-mercaptoethanol and 1 mM PMSF and centrifuged as described above. The supernatant was loaded on a pre-equilibrated Hi-Trap Ni-Sepharose column (GE Healthcare). After extensive washing of the column with lysis buffer containing 30 mM imidazole but lacking PMSF, bound protein was eluted in a linear 30–300 mM imidazole gradient over 30 column volumes. Protein-containing fractions were pooled and dialyzed against buffer containing 50 mM HEPES, pH 7.5, 150 mM NaCl, and 1 mM $MgCl_2$. The buffer was exchanged twice.

(3) HOPS, yEGFP-fused HOPS and Mon1-Ccz1. HOPS, yEGFP-fused HOPS and Mon1-Ccz1 were expressed in *Saccharomyces cerevisiae* and purified essentially as described before (*Lürick et al., 2017*). For yEGFP-fused HOPS, yEGFP was fused to the C-terminus of Vps proteins via a linker of RTLNVDGSGAGAGAGAGAIL. yEGFP-fused HOPS and Mon1-Ccz1 complexes were expressed in *S. cerevisiae*. In short, cells were grown until an $OD_{600}$ of around 8. Harvested cells were resuspended in 50 mM HEPES, pH 7.4, 300 mM NaCl, 1.5 mM $MgCl_2$, 1 mM DTT, 0.5 mM PMSF, 1x FY protease inhibitor mix (Serva, Germany), and 10% glycerol. For purification of Mon1-Ccz1, the salt concentration was decreased to 150 mM NaCl. Cell lysis in the presence of glass beads was conducted in a FastPrep-24 5G (MP, Germany). After removal of the glass beads, the supernatant was centrifuged at 120,000 g for 1 hr at 4°C. Centrifuged lysate was incubated with pre-equilibrated immunoglobulin G (IgG) Sepharose (GE Healthcare) for 2 hr at 4°C. IgG beads were washed with 15 ml lysis buffer lacking PMSF and FY, and bound proteins were eluted by cleavage with TEV protease for 1 hr at 16°C.

Strains and plasmids used in this study are listed in *Supplementary file 3*.

## In vitro prenylation of Rab GTPases

Rab–GDI complexes were obtained from prenylation reactions (*Thomas and Fromme, 2016*). 10 µM Rab GTPase pre-loaded with GDP was incubated with 9 µM GDI, 1 µM Rab escort protein (REP) Mrs6, 1 µM geranylgeranyl transferase (Bet2-Bet4), and a sixfold excess of geranylgeranyl pyrophosphate in assay buffer (50 mM HEPES, pH 7.5, 150 mM NaCl, 2 mM $MgCl_2$, 1 mM DTT) for 1 hr at 37° C. Mrs6 and Bet2-Bet4 fused to a His6 tag were removed by subsequent incubation of the reaction with Ni-NTA Agarose (Macherey-Nagel, Germany) for 1 hr at 4°C. The stoichiometric complex containing the prenylated Rab and GDI was isolated from the supernatant by size exclusion chromatography. The functionality of prenylated Rabs was tested in membrane association and tethering assays as described before with modifications (*Lürick et al., 2017*).

For membrane association and tethering assays, lipid films were evaporated by a SpeedVac (CHRIST, Germany) and resuspended in HEPES-KOAc (HK) buffer (50 mM HEPES, pH 7.4, and 120 mM KOAc). Liposomes as unilamellar vesicles were obtained by five freeze and thaw cycles in liquid nitrogen. For tethering assays, the 2 mM liposome suspension was extruded through polycarbonate filters (400 nm, 200 nm, and 100 nm pore size) using a hand extruder (Avanti Polar Lipids, Inc). To analyze the membrane association of mNeon-Ypt7, 50 pmole mNeon-Ypt7 complexed with GDI was incubated with 50 nmole liposomes (62 mol % POPC, 18 mol % POPE, 10 mol % POPS, 8 mol % ergosterol, 1 mol % DAG, and 1 mol % PI-3-P) in the presence or absence of GTP for 30 min at 27° C. Liposomes were sedimented by centrifugation for 20 min at 20,000 g. The fraction of membrane-bound mNeon-Ypt7 was determined by the fluorescent signal of the supernatant before and after sedimentation, which was quantified in a SpectraMax M3 fluorescence plate reader (Molecular Devices, Germany).

For analysis of HOPS-mediated tethering, liposomes containing 69 mol % POPC, 18 mol % POPE, 8 mol % ergosterol, 1 mol % DAG, 1 mol % ATTO488-DPPE, and 3 mol % DOGS-NTA or compensatory amounts of POPC were used. Ypt7 C-terminally fused to a His6 tag was loaded with GDP or GTP and afterwards targeted to liposomes via the lipid analogue DOGS-NTA (*Lürick et al., 2017*). Liposomes lacking DOGS-NTA were directly loaded with prenylated Ypt7. In this case, 50 pmole pYpt7:GDI complex was incubated with 50 nmole liposomes in the presence of GTP for 30 min at 27°C. For tethering reactions, 0.170 mM Ypt7-loaded liposomes were incubated with 50–350 nM HOPS complex or buffer, respectively, for 10 min at 27°C. Liposomal clusters were sedimented for 5 min at 1,000 g. The fraction of tethered liposomes in the pellet was determined on the basis of the ATTO488 fluorescent signal in the supernatant before and after sedimentation, using a SpectraMax M3 fluorescence plate reader.

## Coating graphene monolayer on substrates

Silica-type substrates were used for coating with graphene monolayer. These include $1 \times 1$ cm$^2$ silica-coated transducers for TIRFS-RIF detection and glass coverslips for microscopy imaging. Before coating, the substrates were cleaned by plasma cleaner (Femto plasma system, Diener electronic GmbH/Germany). Coating of graphene monolayer on substrate was based on the manufacturer's instruction. Briefly, the Easy Transfer Monolayer containing graphene monolayer with a thin protecting polymer film was cut into $0.5 \times 0.5$ cm$^2$ pieces. The obtained piece was slowly emerged into MilliQ water to float on top of water. The polymer-protected graphene monolayer was fished out by a clean glass coverslip or TIRFS-RIF transducer from below, resulting in face-to-face contact of graphene monolayer with the substrate. The obtained substrate was left drying at room temperature for 30 min, followed by heating in a 150°C oven for 2 hr. The hot substrate was taken out and immediately stored under vacuum for 24 hr for cooling down. With the protecting polymer film, the obtained graphene-coated substrate could be stored at ambient conditions for weeks. Immediately before the experiments, the graphene-coated substrate was incubated in acetone for 1 hr, sequentially in isopropanol for another 1 hr, to remove the protecting polymer film on graphene. By blow-drying with $N_2$ stream, the graphene monolayer-coated substrate was ready for use.

## Preparation of liposomes and formation of solid-supported membranes

For preparation of DOPC liposomes, 2.5 µmol DOPC was dissolved in chloroform in a 50 ml round bottom flask. For preparation of liposomes containing DOPC:DOPS (95:5 molar ratio), 2.4 µmol DOPC and 0.12 µmol DOPS dissolved in chloroform were mixed in a 50 ml round bottom flask.

Similarly, a lipid mixture of 2.4 µmol DOPC and 0.125 µmol trisNTA-DODA was used for preparing liposomes of DOPC:trisNTA-DODA (95:5 molar ratio). Liposomes were prepared as small unilamellar vesicles (SUVs) by probe sonication as described before (*Beutel et al., 2014*). To form solid-supported membranes on glass coverslips, 800 µl of liposome solution was added on top of a freshly cleaned coverslip. After incubation at room temperature for 20 min, the coverslip was rinsed with excess HBS to remove free vesicles. For binding His-tagged proteins on solid-supported membranes of DOPC:trisNTA-DODA, conditioning of trisNTA-DODA was carried out by loading with $Ni^{2+}$ ions. The coverslip was rinsed with 250 mM EDTA and 200 mM imidazole, sequentially. After incubation with 10 mM $NiCl_2$ for 10 min, the coverslip was washed by HBS buffer and 200 mM imidazole to remove possible non-specifically bound $Ni^{2+}$ ions.

## Surface sensitive TIRFS-RIF detection

A home-built set-up for simultaneous total internal reflection fluorescence spectroscopy (TIRFS) (*Gavutis et al., 2006*) and reflectance interference (RIF) detection (*Piehler and Schreiber, 2001*; *Schmitt et al., 1997*) has been described in detail before. For TIRFS-RIF detection of His-tagged protein binding, DOPC/trisNTA-DODA (95:5 molar ratio) was injected into the flow chamber to form lipid bilayers on silica substrate, or lipid monolayer on graphene. Alternatively, solution-assisted lipid deposition (SALD) was used for forming lipid monolayer on graphene, in which ethanol was added to HBS to obtain a final mixture of HBS:EtOH (90:10 v/v). Conditioning of trisNTA-DODA by $Ni^{2+}$ ion loading follows the same protocol as in vitro by using the flow-through system in the TIRFS-RIF setup. On the $Ni^{2+}$-loaded trisNTA-DODA/DOPC lipid mono-/bilayers, 1 µM NB-H6 was injected, followed by injections of 100 nM tagless mEGFP, HBS buffer rinsing and imidazole washing for surface regeneration. For direct immobilization of H6-mEGFP, 1 µM H6-mEGFP was injected followed by imidazole washing. Mass signals of protein binding and fluorescence signals were recorded simultaneously in RIF channel and TIRFS channel, respectively. The ratio of fluorescence intensity $I_G/I_0$ was normalized to the amount of immobilized mEGFP according to *Equation (1)*.

$$\frac{I_G}{I_0} = \left(\frac{m_0}{m_G}\right)\left(\frac{I_{G\_raw}}{I_{0\_raw}}\right) \tag{1}$$

where $m_0$ and $m_G$ are the mass signals of mEGFP immobilized on silica and graphene, respectively, and $I_{0\_raw}$ and $I_{G\_raw}$ are the fluorescence intensities of mEGFP immobilized on silica and graphene, respectively.

For DNA hybridization, liposomes with 250 µM DOPC/DOPS (95:5 molar ratio) was injected to the surface of TIRFS-RIF transducer to form lipid layers on the solid support. Anchor strands with cholesterol modification were injected for immobilization on the surface. Sequentially, probe strand labeled with FAM was injected for hybridization with the anchor strand. All DNA concentrations were 1 µM. The running buffer was HBS buffer containing 5 mM $Mg^{2+}$ for DNA hybridization (HBS-Mg buffer, 20 mM HEPES, 150 mM NaCl, and 5 mM $MgCl_2$, pH 7.5). Fluorescence ratio of $I_G/I_0$ was normalized to the amount of immobilized DNA according to *Equation (1)*. For calibration experiments using OG488-DHPE, liposomes containing 250 µM DOPC mixed with 0.1% molar ratio OG488-DHPE was used. $I_G/I_0$ was obtained by normalizing to the mass signals according to *Equation (1)*.

Binding of Mon1-Ccz1 to lipid layers was probed as described before (*Cabrera et al., 2014*). 250 µM DOPC was injected to the substrates for formation of lipid monolayers on graphene or lipid bilayers on silica, respectively, followed by injection of 100 nM Mon1-Ccz1.

## Electromagnetic simulation of GIET

For simulating the electrodynamic coupling of the excited fluorophore to graphene, the excited fluorophore was treated as an ideal electric dipole emitter and the graphene as a layer of matter with specific thickness and (complex-valued) bulk refractive index as described before (*Ghosh et al., 2019*). Solving Maxwell's equations of such a system leads to an expression for the emission power, $S(\theta, d)$, of the electric dipole emitter as a function of dipole distance $d$ and orientation (described by the angle $\theta$ between the dipole axis and the vertical axis) to the substrate. Considering the flexible linker and rapid rotation of the fluorophore in biomolecules, $S(\theta, d)$ was averaged over random orientations as a simplified $S(d)$. The relative fluorescence lifetime ($\tau_G/\tau_0$) was calculated as:

$$\frac{\tau_G}{\tau_0} = \frac{S_0}{\left(1-\phi\right)S_0 + \phi S\left(d\right)} \tag{2}$$

where $\phi$ is the quantum yield (QY), $\tau_0$ is the free-space lifetime in the absence of GIET, $S_0$ is the free-space emission power of an ideal electric dipole emitter, $S_0 = cnk_0^4 p^2/3$, with $c$ being the vacuum speed of light, $k_0$ is the wave vector in vacuum, $n$ is the refractive index of water ($n$ = 1.33), and $p$ is the amplitude of the dipole moment vector.

The $\tau_G/\tau_0$ as a function of distance $d$ was calculated for the following geometry: glass substrate ($n$ = 1.52) covered with single sheet graphene (thickness = 0.34 nm, $n$ = 2.76 + 1.40$i$ for emission at 670 nm, or $n$ = 2.68 + 1.21$i$ for emission at 520 nm) coated with lipid monolayer ($n$ = 1.44, thickness = 2.5 nm), topped with water ($n$ = 1.33). The emission maximum, QY, and $\tau_0$ of the fluorophores were: FAM (518 nm, 0.75, 3.0 ns), EGFP (507 nm, 0.60, 2.1 ns), mNeonGreen (517 nm, 0.80, 2.8 ns), and Dy647P1 (667 nm, 0.27, 1.3 ns). The emission maximum and QY were taken from the literature (*Mujumdar et al., 1993*; *Shaner et al., 2013*; *Zhang et al., 2014*), values of $\tau_0$ were measured on glass-supported lipid bilayer in this work. Results of distance dependency and sensitivity analysis are summarized in *Figure 2—figure supplement 2*. In the absence of static quenching, fluorescence intensity of fluorophore is proportional to its lifetime. Thus the relative fluorescence intensity follows the same theoretical prediction based on *Equation (2)*.

## Validation of distance-dependent GIET by DNA nanorulers

To calibrate the distance-dependent GIET, the measured intensity ratios and lifetime ratios of DNA nanorulers need to be plotted vs the vertical distance of the FAM dye above graphene. The vertical distance $d$ was calculated by considering tilting of the DNA strands on surface (*Wong and Pettitt, 2004*):

$$d = l_{DNA} * sin\alpha + l_{ML} \tag{3}$$

where $l_{DNA}$ is the end-to-end length of FAM dye on the hybridized probe strand to the 3′ end of the anchor strand. $l_{DNA}$ = 1.7, 5.1, 6.8, 8.5, 10.2, 11.9, and 17.0 nm. It is set as 0 nm for OG488-DHPE. $\alpha$ is the tilting angle between DNA and graphene surface. $l_{ML}$ = 2.5 nm, is the thickness of lipid monolayer on graphene. The anchor DNA strands were modified with a cholesterol at 3′ end via a tri-ethylene glycol phosphate linker ('Spacer 9' of Integrated DNA Technologies, Inc). A full integration of the anchoring group into the phospholipid monolayer was assumed. Using the '*lsqcurvefit*' function in Matlab, a global fitting of the intensity or lifetime ratios vs $d$ to the simulated GIET curve of FAM yielded $\alpha$ of 42° (intensity) or 44° (lifetime).

## Confocal laser-scanning microscopy and lifetime measurements

Lifetime measurements were carried out on a confocal laser-scanning microscope (FluoView 1000, Olympus) equipped with a FLIM/FCS upgrade kit from PicoQuant using a 60× (NA 1.2) water-immersion objective (UPLSAPO, Olympus). EGFP/mNeonGreen was excited either by the 488 nm line of an argon laser (Olympus) for cLSM/FRAP or by a picosecond pulsed 485 nm laser diode at 40 MHz repetition rate (LDH-D-C-485, Picoquant). Time-correlated single photon counting (TCSPC) was performed using the TCSPC module PicoHarp 300 (PicoQuant together with a picosecond laser driver Sepia II (PicoQuant) and a single photon avalanche detector (PicoQuant)). Emission photons were filtered by a 500–550 nm bandpass filter (BrightLine HC 525/50, Semrock). TCSPC was acquired using point measurement or image mode (i.e. FLIM mode). Acquisition time was more than 90 s to obtain >$10^5$ counts per sample for robust lifetime analyses. If not mentioned elsewhere, the TCSPC histograms were tail-fit with mono-exponential decay function using SymPhoTime64 integrated in the PicoQuant system. Only for mNeon-Ypt7 on graphene, the TCSPC histograms were fit with bi-exponential decay functions by fixing one component to the obtained average lifetime of mNeon-Ypt7 on glass (<15%).

## Fluorescence lifetime and FRAP of DNA strands

800 µl 250 µM DOPC/DOPS (95:5 molar ratio) was added on top of graphene-coated glass coverslips. After incubation at room temperature for 5 min, the coverslips were washed for 5 times with 1

ml HBS-Mg buffer. The cholesterol-modified ssDNA anchor strands were added in the solution with a final concentration of 200 nM. After 5 min incubation, excess anchor strands were removed by washing for 5 times with HBS-Mg buffer. Probe strands of FAM-labeled ssDNA were added with a final concentration of 200 nM. After 5 min incubation, excess probe strands were removed by washing for 10 times with HBS buffer. For fluorescence lifetime measurements, a final concentration of 10 µM EDTA was added to the solution for preventing possible metal ion-mediated fluorescence quenching. TCSPC was acquired by point measurement mode. Lifetimes were obtained by monoexponential fitting of TCSPC curves. For FRAP experiments of DNA strands, a region of interest (ROI) was placed in areas with and without graphene, respectively.

In FRAP experiments of DNA strands, a circular region of interest (ROI) with a radius of 10 µm was selected. Five pre-bleach images were recorded before the ROI was illuminated by 405 nm in 10 s. A background signal $I_{BG}$ was recorded from the center of the ROI immediately after photobleaching. Fluorescence recovery was monitored at a time resolution of 0.8 s for glass or 3.2 s for graphene. The fluorescence intensity in the ROI ($I_{ROI}$) was obtained by subtracting the background. A reference region outside the photobleached ROI was processed in the same way to obtain $I_{REF}$. ROI intensity was normalized to the reference ($I_{ROI}/I_{REF}$) for each time interval using *Equation (4)*:

$$\frac{I_{ROI}}{I_{REF}} = \frac{I_{ROI\_inside} - I_{BG}}{I_{ROI\_outside} - I_{BG}} \tag{4}$$

where $I_{BG}$ is the background signal, $I_{ROI\_inside}$ is the fluorescence intensity within the ROI, and $I_{ROI\_outside}$ is the intensity outside the photobleaching ROI. Plotting $I_{ROI}/I_{REF}$ vs time yields the time-lapse FRAP curve.

## FLIM and FRAP of mNeon-pYpt7 and yEGFP-fused HOPS

Graphene-coated glass coverslips were incubated with 800 µl DOPC and washed as described above. Membrane association of 200 nM prenylated mNeon-Ypt7 or 150 nM prenylated Ypt7 complexed with GDI was conducted in the presence of 20 mM EDTA, pH 8.0 and 1 mM GTP for 30 min at 30°C. The loading reaction was stopped by addition of 25 mM MgCl$_2$ and incubation for 10 min at room temperature. The coverslip was washed extensively with HBS (>15 times) to remove excess EDTA and Mg$^{2+}$ ions. Where indicated, 100 nM Mon1-Ccz1 or HOPS were added to mNeon-pYpt7-GTP-loaded membranes for 5 min at room temperature. Before fluorescence lifetime determination, the unbound complexes in solution were removed by washing for 8 times with HBS. For loading of HOPS complexes containing yEGFP-fused subunits on membranes, 50 nM HOPS carrying one subunit (Vps39, Vps11, Vps18, Vps33 or Vps16) fused to yEGFP was added to pYpt7-GTP-loaded membranes for 5 min at room temperature. Before fluorescence lifetime determination, the coverslip was washed 5 times with HBS to remove unbound HOPS in solution. Fluorescent lifetimes of mNeon and yEGFP, respectively, were determined by TCSPC acquired in image mode (FLIM). Fluorescence lifetime ratios were determined based on lifetimes obtained on graphene and glass, respectively, and the distance from the graphene surface $d$ were calculated based on the calibration curve (*Supplementary file 2*). The distances from the monolayer surface $h$ were calculated from $d$ by subtracting the thickness of lipid monolayer (2.5 nm).

For FRAP experiments of mNeon-pYpt7-GTP and mNeon-pYpt7-GTP in complex with Mon1-Ccz1 and HOPS, respectively, a ROI with a radius of 7.5 µm was illuminated with a 405 nm laser for 8.2 s. The fluorescence recovery was monitored for 72 s at a time resolution of 1.8 s (excitation: 488 nm). The $I_{ROI}/I_{REF}$ for each time interval was obtained using *Equation (4)* as described above. All FRAP experiments were carried out at room temperature.

## Single-molecule fluorescence microscopy

Single-molecule imaging experiments were conducted by total internal reflection fluorescence (TIRF) microscopy with an inverted microscope (Olympus IX71) equipped with a triple-line total internal reflection (TIR) illumination condenser (Olympus) and a back-illuminated electron multiplying (EM) CCD camera (iXon DU897D, Andor Technology) as described before (*Wilmes et al., 2020*). A 150 × magnification objective (UAPO 150×/NA 1.45 TIRFM, Olympus) was used for TIR illumination resulting in an image pixel size of 107 nm. Image acquisition was performed at 30 frames per second using an exposure time of 32 ms per frame. All experiments were carried out at room temperature

in presence of oxygen scavenger, that is 0.5 mg/ml glucose oxidase, 0.04 mg/ml catalase, 5% w/v glucose. 150 nM pYpt7-GTP was loaded onto glass-supported lipid bilayers and graphene-supported lipid monolayers for 30 min at 30°C. 50 nM HOPS with Vps33 or Vps11 fused to yEGFP was added in solution, respectively, followed by 10 times buffer washing. To ensure reliable single-molecule detection, 50 pM of dye-labeled NB was used to label the yEGFP-fused HOPS complex. The typical density of NB-labeled HOPS was ~0.2–1 molecule/$\mu m^2$.

### Single-molecule photobleaching

Photobleaching steps were counted manually from time-lapse single-molecule intensity traces as before (*Wedeking et al., 2015b*). The single-molecule intensity traces were obtained by using ImageJ software, in which a 5 × 5 pixel region of interest (sROI) was used for cropping individual molecules. More than 100 intensity traces were screened for counting the discrete steps for each sample (*Figure 5—figure supplement 1*).

### Single-molecule intensity analysis and hidden Markov modeling

Single emitters in time-lapse TIRFM images were localized via a 2D Gaussian mask (*Arnauld et al., 2008*; *Thompson et al., 2002*). Intensity-time traces were built by grouping localizations within a 150 nm search radius of each other with a minimum observation time of 100 frames as described before (*Niewidok et al., 2018*). In order to exclude crosstalk between traces only identified immobilization events with a minimum center-to-center distance of 500 nm apart were selected for further analysis. Individual traces were initially analyzed and evaluated for goodness-of-fit using the step transition and state identification (STaSI) algorithm (*Shuang et al., 2014*; *Wedeking et al., 2015b*). The best fitting 33% of these traces were used to estimate optimal initial values (initial state distribution, transition matrix and state emission functions) for subsequent training of a Hidden Markov Model (HMM) with Gaussian state emissions (*Rabiner, 1989*). To the end, the most probable state sequence (Viterbi path) was calculated for each trace by assigning a state to each single-molecule localization. The relative state occupations were calculated as the number of localizations classified into the specific state with respect to all observations. State transition rates were calculated by estimating the dwell time of each state using a mono-exponential fit of identified state segments as well as the number of observed state transitions according to the method described before (*Yang et al., 2018*). The state definition, occupancy and transition rates were summarized in *Supplementary file 4*.

### Axial localization precision of smGIET

The axial localization precision of smGIET was determined by comparing the root mean square deviation (RMSD) with the mean intensity of individual molecules. To avoid intensity fluctuations on graphene due to GIET, RMSD and mean intensities were quantified based on single-molecule detections on glass coverslips (*Figure 5—figure supplement 3*). For intensities in the range of 300–700 a.u., the obtained ratios of RMSD to mean were very similar (*Supplementary file 5*). Based on the ratios, a relative error of 7.5 ± 0.5% was obtained for single-molecule detections. The axial localization precision depends on $I_G/I_0$, which has a propagated relative error of $\sqrt{2} \times 7.5\%$ = 10.6%. Thus, the relative single-molecule axial localization precision was 10.6% of the corresponding distance. Given the sensitive range of GIET in 3–30 nm, the corresponding axial precision is in the range of 0.3–3 nm.

### Data and materials availability

All data needed to evaluate the conclusions in the paper are present in the paper and/or the Supplementary Materials. Additional data related to this paper may be requested from the authors.

## Acknowledgements

The authors thank Anna Lürick and Cornelia Bröcker for HOPS characterizations, and Gabriele Hikade, Hella Kenneweg, Kathrin Auffarth, and Angela Perz for expert technical assistance. Funding: The work was funded by the DFG (SFB 944 P8, SFB 944 P11, SFB 944 Z, INST 190/182–1, UN 111/12-1, and YO 166/1-1), NSFC (31870976) and by intramural funding from Osnabrück University

within the profile line 'Integrated Science'. JE acknowledges support by the DFG under Germany's Excellence Strategy (EXC 2067/1- 390729940).

## Additional information

### Competing interests

Jacob Piehler, Changjiang You: Inventor on a patent application submitted by University of Osnabrück that covers the use of GIET for biomolecule detection (EP 20 175 412.4). The other authors declare that no competing interests exist.

### Funding

| Funder | Grant reference number | Author |
|---|---|---|
| Deutsche Forschungsgemeinschaft | INST 190/146-3 (SFB944-P8) | Jacob Piehler |
| National Natural Science Foundation of China | 31870976 | Zehao Li<br>Changyuan Yu |
| Intramural funding from Osnabrueck University | Integrated Science | Carola Meyer<br>Christian Ungermann<br>Jacob Piehler<br>Changjiang You |
| Deutsche Forschungsgemeinschaft | EXC 2067/1- 390729940 | Jörg Enderlein |
| Deutsche Forschungsgemeinschaft | INST 190/182-1 | Rainer Kurre<br>Jacob Piehler |
| Deutsche Forschungsgemeinschaft | INST 190/149-3 (SFB944-P11) | Christian Ungermann |
| Deutsche Forschungsgemeinschaft | INST190/152-3 (SFB944-Z) | Jacob Piehler |
| Deutsche Forschungsgemeinschaft | UN 111/12-1 | Christian Ungermann |
| Deutsche Forschungsgemeinschaft | YO 166/1-1 | Changjiang You |

The funders had no role in study design, data collection and interpretation, or the decision to submit the work for publication.

### Author contributions

Nadia Füllbrunn, Data curation, Formal analysis, Validation, Investigation, Visualization, Methodology, Writing - original draft, Writing - review and editing; Zehao Li, Data curation, Formal analysis, Validation, Investigation, Visualization, Methodology, Writing - original draft; Lara Jorde, Data curation, Methodology; Christian P Richter, Software, Validation, Investigation, Visualization, Writing - original draft; Rainer Kurre, Formal analysis, Funding acquisition, Validation, Visualization, Methodology, Writing - review and editing; Lars Langemeyer, Data curation, Investigation, Methodology; Changyuan Yu, Funding acquisition, Investigation, Methodology; Carola Meyer, Conceptualization, Funding acquisition, Investigation, Methodology, Writing - review and editing; Jörg Enderlein, Conceptualization, Software, Formal analysis, Visualization, Methodology, Writing - review and editing; Christian Ungermann, Jacob Piehler, Conceptualization, Formal analysis, Supervision, Funding acquisition, Investigation, Visualization, Methodology, Writing - original draft, Writing - review and editing; Changjiang You, Conceptualization, Formal analysis, Supervision, Funding acquisition, Investigation, Visualization, Methodology, Writing - original draft, Project administration, Writing - review and editing

## Author ORCIDs

Lars Langemeyer ![ORCID] http://orcid.org/0000-0002-4309-0910
Carola Meyer ![ORCID] http://orcid.org/0000-0003-0851-2767
Jörg Enderlein ![ORCID] http://orcid.org/0000-0001-5091-7157
Christian Ungermann ![ORCID] https://orcid.org/0000-0003-4331-8695
Jacob Piehler ![ORCID] https://orcid.org/0000-0002-2143-2270
Changjiang You ![ORCID] https://orcid.org/0000-0002-7839-6397

## Decision letter and Author response

Decision letter https://doi.org/10.7554/eLife.62501.sa1
Author response https://doi.org/10.7554/eLife.62501.sa2

## Additional files

### Supplementary files
- Supplementary file 1. GIET calibration by DNA nanorulers.
- Supplementary file 2. Fluorescence lifetime ratios of endocytic proteins and protein complexes.
- Supplementary file 3. Plasmids and yeast strains used in this study.
- Supplementary file 4. Conformational states and transition kinetics obtained from smGIET.
- Supplementary file 5. RMSD and mean intensities of single-molecule detections.
- Transparent reporting form

### Data availability

All data generated or analysed during this study are included in the manuscript and supporting files.

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
