## [Decision Letter]

**Acceptance summary:**

The authors use graphene-induced energy transfer as a unique spectroscopic ruler to reveal the axial orientation and dynamics of macromolecular complexes in biological membranes with sub-nanometer resolution. This development is a very significant contribution that can help in future mechanistic studies of numerous key biological processes. To illustrate this approach, the authors determine the orientation of the Rab7-like GTPase Ypt7 and of the multi-protein HOPS tethering complex at a lipid monolayer surface.

**Decision letter after peer review:**

Thank you for submitting your article "Nanoscopic anatomy of dynamic multi-protein complexes at membranes resolved by graphene induced energy transfer" for consideration by *eLife*. Your article has been reviewed by three peer reviewers, and the evaluation has been overseen by Felix Campelo as Reviewing Editor and José Faraldo-Gómez as Senior Editor. The following individual involved in review of your submission has agreed to reveal their identity: Volker Kiessling (Reviewer #1).

The reviewers have discussed the reviews with one another and with the Reviewing Editor. Based on these discussions, we have concluded that a revised version of your manuscript might be suitable for publication in *eLife*. The Editors has drafted this decision letter to help you prepare a revised submission.

We would like to draw your attention to changes in our revision policy made in response to COVID-19 (https://elifesciences.org/articles/57162). Specifically, when editors judge that a submitted work as a whole belongs in *eLife* but that some conclusions require a modest amount of additional new data, as they do with your paper, we are asking that the manuscript be revised to either (a) include the new data, if this is possible; otherwise, (b) limit claims to those supported by data in hand; or (c) explicitly state that the relevant conclusions require additional supporting data. In the latter case, our expectation is that the authors will eventually carry out the additional experiments and report on how they affect the relevant conclusions either in a preprint on bioRxiv or medRxiv, or if appropriate, as a Research Advance in *eLife*, either of which would be linked to the original paper.

Summary

The authors use GIET as a unique spectroscopic ruler to reveal the axial orientation and dynamics of macromolecular complexes at biological membranes with sub nanometer resolution, which is a very significant contribution that can help in the future mechanistic understanding of numerous key biological processes. GIET is based on the fluorescence quenching in close proximity to a graphene surface. The system is calibrated by utilizing "DNA nanorulers" of different length and with fluorescent labels at different specific positions. Measuring the fluorescence intensities and fluorescence lifetimes relative to unquenched samples. The calibrated system was then used to determine the orientation of the Rab7-like GTPase Ypt7 and of the multiprotein HOPS tethering complex at a lipid monolayer surface. The normal distances of specific residues to the monolayer surface are determined in ensemble measurements and the transition between three distinguishable conformations are measured in single molecule imaging experiments.

All three reviewers highlighted the importance and the quality of the research presented in this manuscript. However, the reviewers also raised important questions and concerns, detailed below, which must be satisfactorily addressed before the manuscript can be accepted for publication. Given that this manuscript is being considered for a "Tools and Resources" article, it will be crucial for the authors to carefully address the reviewers' comments to further establish the validity and scope of the methodology and the clarity of the manuscript.

In particular, it will be important to fully clarify the results shown in Figure 2 (distance calibration with the nanorulers), in the context of reviewer #2 comments. It will also be important that the authors discuss in more detail the limitations of their technique and the expected errors in the measurements. The authors are also very strongly encouraged to perform the additional control experiments suggested by the reviewers, which appear relatively straightforward and not very time-consuming, as they would greatly strengthen the significance and impact of the study.

Essential revisions

Reviewer #1:

Nadia Fuellbrunn and co-workers present their work about a new fluorescence method to determine the conformation and its dynamic of protein complexes by measuring the normal distance of site-specific fluorophores relative to a lipid monolayer. The graphene induced energy transfer (GIET) assay is based on the fluorescence quenching in close proximity to a graphene surface. The system is calibrated by utilizing "DNA nanorulers" of different length and with fluorescent labels at different specific positions. Measuring the fluorescence intensities and fluorescence lifetimes relative to unquenched samples. The calibrated system was then used to determine the orientation of the Rab7-like GTPase Ypt7 and of the multiprotein HOPS tethering complex at a lipid monolayer surface. Both proteins are part of the yeast vacuolar fusion and tethering machinery. The normal distances of specific residues to the monolayer surface are determined in ensemble measurements and the transition between three distinguishable conformations are measured in single molecule imaging experiments.

The authors present a very nice and useful new method with a lot of potential in particular if it could be used to follow conformational changes of membrane proteins. The approach of measuring the normal distance of site-specific labeled residues is very similar to the method of site-specific fluorescence interference microscopy (sdFLIC, e.g. Nature structural and molecular biology 25 (10), 911), where the conformation of SNARE proteins reconstituted in supported membranes had been investigated. Although the overall strategy seems very similar, sdFLIC hasn't been mentioned in the Introduction where the authors discuss several methods by which protein conformations in membranes can be investigated.

The authors clearly show the usefulness of this new method especially in the dynamic measurements on single molecules. However, there are a couple of open questions and issues that should be addressed and clarified:

A lipid monolayer is not a membrane and the authors should be clear and consistent in their writing in the whole manuscript (e.g. “…nanometer sensitivity of quantifying the axial position of membrane-bound proteins by GIET, 215.how the functional complex is organized on membranes,…”). Proteins are not reconstituted into a membrane on graphene. The fact that the HOPS complex clusters in a diffusible membrane shows that this situation (and potentially the conformation) is very different from the one at the monolayer. This raises another question. Is it justifiable to use the membrane system to measure the unquenched intensities/lifetimes to normalize the monolayer data? Shouldn't this be done on a monolayer too?

A thorough and extensive error discussion is missing. Since this work is presented as a first application of this method, a detailed discussion and quantification of all error sources (statistical and systematic) is necessary. One would expect these errors to be different for distance changes of one fluorophore compared to the errors of absolute distances.

The data points in the calibration Figure 2B are already deviating from the fit curve especially in the range between 3 and 8 nm. This should be quantified and reported. Wouldn't this be a minimum systematic error? The deviation looks systematic, so there might be something missing in the theory? Could a fluorescein labeled lipid be used to determine the 0 nm (2.5 nm from graphene) point in the calibration curve?

Related to the previous point. Without error discussion can the absolute distances really be reported with Å accuracy as in Figures 3 and 4.

Could some of the observed distance changes be due to orientation changes of the dipole? The theoretical curve assumes isotropic orientation. Is this justified under all conditions (different conformations, binding of HOPS to Ypt7)?

In ensemble experiments where a distance change upon binding of a ligand is observed like in Figure 2D or in Figure 3G, the authors should show the specificity and concentration dependence of the result. Titrating the ligand should confirm that the reaction and with that the observed distance changes are in saturation. A comparison with a non-specific membrane binding protein or protein domain should confirm the specificity of the observed distance change on pYpt7.

Related to this issue: How reproducible are the pYpt7 densities in the monolayer or membrane? If ligand binding is in saturation small density differences might not matter, but the authors should show that.

Figure 3—figure supplement 4 shows that the measured lifetimes of mNeon-pYpt7 in membranes on glass are split into two populations. This is not discussed, and it looks like the mean value of the two components was used to normalize the graphene data. The graphene data either doesn't show or resolve different populations on a monolayer.

Similar to the glass data above, the graphene data of Vps33-yEGFP in Figure 4—figure supplement 2 shows two populations (possibly even Vps39). The mean of the two doesn't seem to represent any data points but it was used for the result. Please comment and discuss.

The statistics of the results are not clear. How many measurements were performed per sample, how many independently prepared samples were used?

Reviewer #2:

This is a well-written and comprehensive manuscript describing a novel methodological approach to probe axial organization of proteins labeled with fluorescence dyes in the nanometer range and in the context of membranes. For this, the authors developed a lipid monolayer assembly on graphene for site-specific protein capturing. Graphene-induced energy transfer (GIET) was adapted by replacing metal with graphene, an optical transparent material. GIET has advantages analogous to FRET as well as TIRF, such as a strong distance dependence and the ability to measure nanometer-range distances. The specific properties of graphene lead to a ~30 nm axial resolution, according to the author's calculations. The authors first validated GIET's ability to measure distance within a dynamic range of 25 nm on graphene-supported lipid monolayers using DNA oligonucleotides as a nanoscale ruler. The authors applied this approach to dissecting the axial organization and dynamics of HOPS-mediated large multi-protein complex regulated by Rab7-like Ypt7, involved in vesicular transport and fusion.

Strengths of this manuscript are the comprehensive nature of the methodological approach described. The authors provide validation of GMIET distance-based measurements. The authors indicate that HOPS complex adopts an upright orientation on membranes with interesting axial dynamics. Their results introduce GIET to significant biological problems as a powerful technique to characterize nanoscale spatiotemporal organization of multi protein complexes at membranes. Another strength of this paper is the ability to use GIET distance measurements to monitor conformational transitions dynamics, e.g. "breathing" of the HOPS complex.

Weaknesses are inherent to the probing of higher-order complexes at membranes using a complex methodology like GMIET, which requires taking those complexes out of their biological membrane background for the analysis. In fact, that contradiction reveals itself in the Introduction, where the authors first mention that "mechanistic understanding of large membrane protein complexes has been addressed by techniques […] that […] take them out of their biological membrane context". Later the authors similarly propose to use distance dependent fluorescence quenching by graphene as a novel technique to quantify the axial organization and dynamics of large membrane protein complexes upon taking them out of their native membrane context. It would be good for the authors to accept that their methodology does indeed provide an improvement in the ability to provide spatiotemporal solution in the 20-30nm but still works within a reconstructed system framework that as some of the other experimental approaches that the authors criticize, attempts to reconstruct large membrane complexes in an artificial lipid monolayer and thus out of their native environment and thus out of context.

Nevertheless, this is a tour de force by the authors to completely reconstruct the HOPS-complex regulated by Rab7-like Ypt7, to highlight the value of the GIET technique to measure distances between fluorophore and graphene layer.

A few comments should be addressed:

– Figure 2 is the core of the paper and the ability to validate the distances of the ssDNA designed as nanorulers sustains all the remaining results. If the results in Figure 2 are questionable, then all other results would be problematic. So, it is important to question why the authors need to include undefined parameters to match their results to the ground truth distance measurements provided by the DNA nanoruler system. The consideration of the tilting parameter is unfortunate and would be important to find a system in which ground truth can be measured without data processing and calculation to match model to experimental data.

– Also is not clear why the authors rely on the ratio between the measurements generated from graphene vs silica layers?

There should be a theoretical explanation for these statements:

– "Strikingly, the obtained correlation of IG/I0 vs d matches the predicted GIET curve well with a globally fitted tilting angle of (43 ±1)o (Figure 2B), confirming the distance-dependent GIET on graphene-supported lipid monolayer."

– "Ratios of fluorescence lifetimes on graphene (τG) to those on glass (τ0) closely matched the corresponding intensity ratios, which also confirmed the simulated distance dependent GIET efficiency (Figure 2B, Supplementary file 1C)".

– Similarly, for longer distances >10nm it would be important to have a DNA ruler than could directly be measured as ground truth using GIET on graphene-supported lipid monolayer without needing comparison with silica substrate measurements.

– Also it is interesting that a similar angle of 45o is found for calibration curve as well as for the measurements of 35mer DNA nanoruler: "Comparing to the length of the 35mer DNA nanoruler (11.9 nm), the height corresponds to a tilting angle α = 45{degree sign}, which is in good agreement with the tilting angle α obtained for the calibration." Is there an explanation for this finding/coincidence?

Reviewer #3:

In brief, this work highlights GIET as a unique spectroscopic ruler to reveal the axial orientation and dynamics of macromolecular complexes at biological membranes with sub nanometer resolution, which is a very significant contribution to the mechanistic understanding of numerous key biological processes.

Surface plasmon enhancement of electromagnetic field on metals, semiconductors and graphene has been implemented in various biosensing applications, covering a spectral region from UV-visible to IR. Comparing to the broad applications of surface enhancement, the nearfield radiationless electromagnetic coupling between the surface plasmon and excited fluorephores is less harnessed for biosensing. The situation is changing. Metal induced energy transfer (MIET) and graphene-based MIET (equivalent to “GIET” of graphene induced energy transfer) have emerged recently. In this manuscript, Fullbrunn et al. have set up the first step of using GIET in the field of protein structural biology. In contrast of detecting the structures of DNA origami or likely robust antibody-based biomolecular analysis, the current work draws a high standard by using lipidated membrane association protein and a tricky multiprotein complex. The authors' strategy is straightforward which achieved their goals efficiently: biofunctionalizing graphene on glass, validating the distance dependent GIET, and determining the structure of protein complexes. With the interdisciplinary expertise, the authors successfully used GIET as a spectroscopic ruler mapping the axial distance of the ~20 nm long HOPS complex on graphene. Very impressively, the structural dynamics of HOPS complexes was quantified by single molecule GIET on graphene-coated microscopy coverslips. I found this research was very well designed and carried out carefully. The manuscript was written in a good logical flow. The conclusions were supported by sufficient figures, tables and their supplementary.

The longer sensitive range of GIET than FRET is indeed very suitable to obtain the structural overview of many extended proteins. As demonstrated in this work, GIET could be a powerful tool for determining the structure of membrane proteins. However, being a surface plasmon-based method, GIET is limited by immobilizing target proteins on the surface. From this aspect, one must point out GIET is not as feasible as FRET for applications in solution. Another disadvantage of the current method is that the coating on graphene is a non-fluidic lipid monolayer (See FRAP experiments in subsection “Axial reorganization at the lipid monolayer surface can be quantified by GIET” and Figure 2—figure supplement 5, Figure 3—figure supplement 4). The authors has noticed in the conclusions “GIET will allow to dissect.… large protein complexes on membranes upon interaction with membrane-bound ligands, and thus test molecular hypotheses in real time”. Actually, instead of considering “interaction of membrane-bound ligands”, surface architectures of fluidic lipid bilayer will be urgently needed for the applications of a large number of biologically important transmembrane proteins.

---

## [Author Response]

Essential revisionsReviewer #1:Nadia Fuellbrunn and co-workers present their work about a new fluorescence method to determine the conformation and its dynamic of protein complexes by measuring the normal distance of site-specific fluorophores relative to a lipid monolayer. The graphene induced energy transfer (GIET) assay is based on the fluorescence quenching in close proximity to a graphene surface. The system is calibrated by utilizing "DNA nanorulers" of different length and with fluorescent labels at different specific positions. Measuring the fluorescence intensities and fluorescence lifetimes relative to unquenched samples. The calibrated system was then used to determine the orientation of the Rab7-like GTPase Ypt7 and of the multiprotein HOPS tethering complex at a lipid monolayer surface. Both proteins are part of the yeast vacuolar fusion and tethering machinery. The normal distances of specific residues to the monolayer surface are determined in ensemble measurements and the transition between three distinguishable conformations are measured in single molecule imaging experiments.The authors present a very nice and useful new method with a lot of potential in particular if it could be used to follow conformational changes of membrane proteins. The approach of measuring the normal distance of site-specific labeled residues is very similar to the method of site-specific fluorescence interference microscopy (sdFLIC, e.g. Nature structural and molecular biology 25 (10), 911), where the conformation of SNARE proteins reconstituted in supported membranes had been investigated. Although the overall strategy seems very similar, sdFLIC hasn't been mentioned in the Introduction where the authors discuss several methods by which protein conformations in membranes can be investigated.

The sdFLIC is indeed a very useful method to detect distance changes of proteins on membranes that we, while being aware of the pioneering work, unfortunately neglected to include into our considerations. In the revised discussion, we now included a thorough discussion of sdFLIC including key references.

The authors clearly show the usefulness of this new method especially in the dynamic measurements on single molecules. However, there are a couple of open questions and issues that should be addressed and clarified:A lipid monolayer is not a membrane and the authors should be clear and consistent in their writing in the whole manuscript (e.g. “…nanometer sensitivity of quantifying the axial position of membrane-bound proteins by GIET, 215.how the functional complex is organized on membranes,...”). Proteins are not reconstituted into a membrane on graphene. The fact that the HOPS complex clusters in a diffusible membrane shows that this situation (and potentially the conformation) is very different from the one at the monolayer. This raises another question. Is it justifiable to use the membrane system to measure the unquenched intensities/lifetimes to normalize the monolayer data? Shouldn't this be done on a monolayer too?

We fully agree with the reviewer that the behavior of proteins in a diffusive membrane is likely very different from their behavior on a lipid monolayer. We particularly emphasized this for the HOPS complex “Given the impaired diffusion in the graphene-supported lipid monolayer, however, the conformationally highly dynamic HOPS complex observed in our studies may represent an intermediate state prior to lateral HOPS clustering, which in turn could increase the efficiency of HOPS to tether incoming vesicles”. This description is based on our observations of: (1) identical binding behavior of Ypt7 and Mon1-Ccz1 on diffusive membranes on glass and on a lipid monolayer on graphene (Results of Mon1-Ccz1 binding are shown in Figure 3—figure supplement 4), and (2) pYpt7-specific binding of HOPS on lipid mono- and bilayers (Figure 4—figure supplement 1). These results suggest that HOPS maintains its functionality and the conformation on both the diffusive membrane and on a lipid monolayer.

We have carefully revised the manuscript to only use the term “membrane” in case of lipid bilayers and otherwise stick to the term “lipid monolayer”. Particularly, the description has been changed to “quantifying the axial position of proteins onto lipid monolayers by GIET” in the revised manuscript.

Regarding the use of the unquenched intensities/lifetimes on a silica-supported membrane for normalization: the lipid composition (e.g. DOPC, DOPS) is the same for lipid mono- and bilayers ensuring very similar dielectric environment that may alter photophysical properties of fluorophores in addition to the effects caused by the molecular context within labeled DNA or proteins. However, these lipids do not absorb visible light and therefore are not expected to affect the fluorescence by inelastic interaction. Therefore, we can safely assume that fluorescence intensity and lifetime are similarly affected on lipid mono- and bilayers. We prefer lipid bilayers for practical reasons, because lipid monolayer formation on silanized, hydrophobic glass surfaces in our hands is much more prone for defect formation, probably due to lack of a perfect hydrophobic silane monolayer.

A thorough and extensive error discussion is missing. Since this work is presented as a first application of this method, a detailed discussion and quantification of all error sources (statistical and systematic) is necessary. One would expect these errors to be different for distance changes of one fluorophore compared to the errors of absolute distances.The data points in the calibration Figure 2B are already deviating from the fit curve especially in the range between 3 and 8 nm. This should be quantified and reported. Wouldn't this be a minimum systematic error? The deviation looks systematic, so there might be something missing in the theory? Could a fluorescein labeled lipid be used to determine the 0 nm (2.5 nm from graphene) point in the calibration curve?

We thank the reviewer for his suggestions. A detailed theoretical error analysis was presented in Figure 2—figure supplement 2: The systematic error is shown in panel B using FAM dye as an example, where the yellow regions mark the variation of *d* within ±5% τ_G_. By plotting the derivative of FAM’s lifetime to distance, a distance dependent sensitivity was obtained, which is presented in panel D. Even though the systematic error increases with distance, accurate distance measurements can still be obtained by increasing the precision of measurement (e.g. reducing the τ_G_ variation as shown by Ghosh et al., 2019). Panel C summarizes the GIET curves of four different fluorophores: FAM, mNeonGreen, EGFP and DY647P1. These curves share a similar distance dependency. Moreover, statistical error of single-molecule GIET was evaluated using the root mean square deviation (RMSD). A relative 10% axial localization precision was obtained for single molecule detection at video rate (Supplementary file 5). Information of these error analyses was mentioned in the manuscript: Similarity of GIET curves for different dyes was described in Discussion paragraph two. Relative axial localization precision for single molecule detection was mentioned in Discussion paragraph one and in Materials and methods (Axial localization precision of smGIET). In the revised manuscript, we now emphasize the results in Figure 2—figure supplement 2 regarding the error analysis in the context of its first citation in subsection “DNA nanoruler calibration confirms distance-dependent GIET efficiency”. We furthermore now cite Figure 2—figure supplement 2 in paragraph two of the Discussion to guide the readers to GIET curves of different fluorophores.

Regarding systematic errors and the deviation of curve fitting in the range of 3-8 nm, we think that this is due to our experiments rather than the theory: (1) The dramatically decreased fluorescence signals at short distance induce errors in measurements. For distances below 5 nm above graphene, less than 10% of the fluorescence signal is left due to strong GIET and therefore potential background signals have stronger bias to both intensity and lifetime measurements. (2) Several flexible linkers are involved in our nanoruler system, which allow increased distances of the fluorophores. Since the fluorophores and the lipid monolayer surface are negatively charged, strong electrostatic repulsion can be expected when fluorophores are dragged into close proximity of the lipid headgroups. Therefore, we took up the reviewer’s excellent suggestion and performed an additional GIET calibration with OG488-DHPE (2′,7′‐difluoro-fluorescein conjugated with 1,2-dihexadecanoyl-*sn*-glycero-3-phosphoethanolamine) integrated into the lipid monolayer. OG488 and fluorescein share largely identical excitation/emission spectra, yielding very similar calculated *I_G_*/*I*_0_ for the “zero position” measurement (2.5% for FAM and 2.4% for OG488). However, we obtained *I_G_*/*I*_0_ = 3.8 ± 0.2% for OG488-DHPE (see Figure 2—figure supplement 1) which is in line with the tendency of systematically decreased GIET-efficiency close to the lipid monolayer surface. Reliable determination of the fluorescence lifetime was not possible for OG488-DHPE integrated into graphene-supported lipid monolayers (Author response image 1,B). While sufficient number of photons could be collected upon extended acquisition, the close overlap with the instrumental response function (IRF) prohibited deconvolution of the curve. This is in line with a predicted fluorescence lifetime below 100ps, which cannot be reliable quantified by our TCSPC system.

**Author response image 1. sa2fig1:** Time-correlated single photon counting (TCSPC) for quantification of GIET efficiency on lipid monolayer. (A) TCSPC curves of OG488-DHPE recorded on glass with 30s (blue), on graphene with 30 s (green) and on graphene with 10 min (red). (B) Zoom up of TCSPC curves obtained on graphene with 30 s (green) and with 10 min (red). (C) Logarithmic diagram of the normalized time-correlated counts obtained on glass 30 s (blue) and graphene 10 min (red), in comparison with the instrument response function (IRF, black).

In the revised manuscript, we have now added the experimental details of the additional experiment in Materials and methods. The intensity measurement for OG488-DHPE have been included into Figure 2B and Supplementary files.

Related to the previous point. Without error discussion can the absolute distances really be reported with Å accuracy as in Figures 3 and 4.

While there is indeed a residual uncertainty concerning the absolute distances, we have very good evidence that the theoretical curve correctly reflects the distance-dependence of GIET, as our electromagnetic calculation of GIET has been previously validated (Ghosh et al., 2019). Under this assumption, we consider it being justified to provide distance data based on the experimental errors of the measurements. For the results of mNeon-Ypt7 (Figure 3), the standard deviation (s.d.) is less than 2%. For HOPS-Vps-yEGFP (Figure 4), the s.d. is less than 4.4%. All the distances of proteins were calculated by considering the s.d. of lifetime measurements. There results were summarized as mean ± s.d. in Supplementary file 2. As shown in Supplementary file 2, the distances can be reported with sub-nanometer precision except for HOPS Vps33-yEGFP.

In the revised manuscript, we have stressed that all experimental distances are based on the assumption that the theoretical curve applies.

Could some of the observed distance changes be due to orientation changes of the dipole? The theoretical curve assumes isotropic orientation. Is this justified under all conditions (different conformations, binding of HOPS to Ypt7)?

Since both yEGFP and mNeonGreen are fused to proteins via long flexible amino acid linkers (12 and 20 residues, respectively), we can confidently assume largely isotropic orientations. While the linker sequence of mNeon-fused Ypt7 was already specified (Materials and methods) we have now also included the linker sequence in yEGFP-fused HOPS subunits. In case of the systems used for calibration – specifically the variants with surface-proximal fluorophore localization – bias by limited flexibility cannot be ruled out entirely. This may also contribute to the systematic deviations observed for these data points.

In ensemble experiments where a distance change upon binding of a ligand is observed like in Figure 2D or in Figure 3G, the authors should show the specificity and concentration dependence of the result. Titrating the ligand should confirm that the reaction and with that the observed distance changes are in saturation. A comparison with a non-specific membrane binding protein or protein domain should confirm the specificity of the observed distance change on pYpt7.Related to this issue: How reproducible are the pYpt7 densities in the monolayer or membrane? If ligand binding is in saturation small density differences might not matter, but the authors should show that.

We agree with the reviewer that it is important to point out that the observed distance changes upon ligand binding are in saturation and specific. We should emphasize that the transfer of pYpt7 to the lipid mono- and bilayers is rather less efficient and thus densities are far below a monolayer and therefore cannot be reliably quantified by our label-free detection. By contrast, binding of Mon1-Ccz1 is very efficient with a density >1 ng/mm² achieved under our experimental conditions (cf. Figure 3—figure supplement 4A), i.e. more than 1/5 of a protein monolayer. This means that Mon1-Ccz1 is available in at least 10-fold excess over pYpt7, thus ensuring pseudo-first order reaction conditions. We actually refrained from further increasing the Mon1-Ccz1 density to prevent bias by crowding the surface.

Since we do not have available another protein that efficiently binds to lipid mono- or bilayer surfaces and we could not use Ni-trisNTA lipids due to His-tagged proteins involved in the reconstitution of pYpt7, we could not perform meaningful negative control experiments. However, we would like to emphasize that Figure 3G itself displays the specificity of observed distance changes for mNeon-Ypt7. Whereas interaction with the guanine nucleotide exchange factor Mon1-Ccz1 leads to an increase in the axial distance by 0.7 nm, the addition of the tethering complex HOPS leads to a decrease by 0.7 nm. Thus, observed distance changes for mNeon-Ypt7-GTP are likely to be related to the function of the ligand and are highly specific.

Figure 3—figure supplement 4 shows that the measured lifetimes of mNeon-pYpt7 in membranes on glass are split into two populations. This is not discussed, and it looks like the mean value of the two components was used to normalize the graphene data. The graphene data either doesn't show or resolve different populations on a monolayer.

This is indeed a somewhat surprising result that may be related to two fluorescent forms of mNeonGreen observed at physiological pH (Steiert, et.al. Biophysical J, 2018): a neutral form (relative quantum yield, QY=1.0, population 60%) and a basic form (relative QY=0.9, population 30%). The population of basic form increases with pH value. The observed two lifetimes in Figure 3—figure supplement 4 have a ratio of 0.9 that matches the previous report. Detection of two lifetimes could be due to changes of the dominant form by heterogeneous surface properties of lipid bilayers on glass. However, the two-population behavior was not repeatedly observed on other glass substrates. We thus treated the two populations as variations in the experiment and used the mean value for normalization. We did not observe two populations of lifetime on graphene, indicating that the possible difference of lifetimes cannot be resolved on graphene. In the revised Figure caption of Figure 3—figure supplement 4, we have noted that mean values of fluorescence lifetimes were used for calculating the ratios.

Similar to the glass data above, the graphene data of Vps33-yEGFP in Figure 4—figure supplement 2 shows two populations (possibly even Vps39). The mean of the two doesn't seem to represent any data points but it was used for the result. Please comment and discuss.

The observation of two populations of the lifetime for Vps33-yEGFP on graphene is indeed intriguing. The situation is different to the abovementioned situation that yEGFP has shown a consistent fluorescence lifetime on glass as shown in Figure 4—figure supplement 2. Most prominently, repeating the experiments of Vps33-yEGFP on graphene resulted in the same two-lifetime profile. Thus, accidental heterogeneity of surface properties can be excluded. Together with the broad lifetime distribution of Vps16-yEGFP shown in Figure 4D, these results may be related to the different HOPS conformations that we identified by single molecule GIET analysis.

In the revised manuscript, we have now specified the reason by choosing HOPS Vps33-yEGFP for single molecule GIET analysis in the revised manuscript (subsection “Structural dynamics of HOPS complex at the membrane”) to emphasize this observation.

The statistics of the results are not clear. How many measurements were performed per sample, how many independently prepared samples were used?

Thank you for pointing this out. Within our manuscript, we describe the sample volumes for all experiments. They are shown in figure captions and notes for tables (also in all supplementary files): see Figures 2, 3, 4 and 5, and their figure supplements; Supplementary files 1, 2, and 4; Table 1, etc. To highlight the robustness of our work, we summarized all information on statistics in *eLife’s* transparent reporting form during submission of the manuscript. With the information now provided in the manuscript and summarized in the transparent reporting form, this information should be provided in sufficient detail.

Reviewer #2:This is a well-written and comprehensive manuscript describing a novel methodological approach to probe axial organization of proteins labeled with fluorescence dyes in the nanometer range and in the context of membranes. For this, the authors developed a lipid monolayer assembly on graphene for site-specific protein capturing. Graphene-induced energy transfer (GIET) was adapted by replacing metal with graphene, an optical transparent material. GIET has advantages analogous to FRET as well as TIRF, such as a strong distance dependence and the ability to measure nanometer-range distances. The specific properties of graphene lead to a ~30 nm axial resolution, according to the author's calculations. The authors first validated GIET's ability to measure distance within a dynamic range of 25 nm on graphene-supported lipid monolayers using DNA oligonucleotides as a nanoscale ruler. The authors applied this approach to dissecting the axial organization and dynamics of HOPS-mediated large multi-protein complex regulated by Rab7-like Ypt7, involved in vesicular transport and fusion.Strengths of this manuscript are the comprehensive nature of the methodological approach described. The authors provide validation of GMIET distance-based measurements. The authors indicate that HOPS complex adopts an upright orientation on membranes with interesting axial dynamics. Their results introduce GIET to significant biological problems as a powerful technique to characterize nanoscale spatiotemporal organization of multi protein complexes at membranes. Another strength of this paper is the ability to use GIET distance measurements to monitor conformational transitions dynamics, e.g. "breathing" of the HOPS complex.Weaknesses are inherent to the probing of higher-order complexes at membranes using a complex methodology like GMIET, which requires taking those complexes out of their biological membrane background for the analysis. In fact, that contradiction reveals itself in the Introduction, where the authors first mention that "mechanistic understanding of large membrane protein complexes has been addressed by techniques… that.… take them out of their biological membrane context". Later the authors similarly propose to use distance dependent fluorescence quenching by graphene as a novel technique to quantify the axial organization and dynamics of large membrane protein complexes upon taking them out of their native membrane context. It would be good for the authors to accept that their methodology does indeed provide an improvement in the ability to provide spatiotemporal solution in the 20-30nm but still works within a reconstructed system framework that as some of the other experimental approaches that the authors criticize, attempts to reconstruct large membrane complexes in an artificial lipid monolayer and thus out of their native environment and thus out of context.Nevertheless, this is a tour de force by the authors to completely reconstruct the HOPS-complex regulated by Rab7-like Ypt7, to highlight the value of the GIET technique to measure distances between fluorophore and graphene layer.

We agree with the reviewer’s insightful comments that our method as currently presented is limited to reconstituting protein complexes after taking them out of their biological membrane context. This approach is similar to traditional high-resolution structural analysis and consequently we aim in the future to combine GIET with single particle transmission electron microscopy using the same graphene supported lipid monolayer system for reconstitution of these complexes on an TEM grid. This will enable powerful correlation of structure and dynamics. Reconstitution furthermore ensures full control on the system with respect to protein and lipid composition and enables sequentially building up the system in situ. In addition, site-specific labeling of purified compounds will in the future enable to interrogate conformational changes much more specifically.

However, our goal is also to extend GIET to analyzing axial organization of membrane proteins in their physiological context, e.g. signaling complexes in the plasma membrane, which are extremely challenging to isolate and reconstitute in a functional manner. We are confident that we can in the future interface cells with glass-supported graphene in a manner that will enable such measurements.

In light of the reviewer’s comment for avoiding possible misleading, we have strengthened the inherent advantage of exploring heterogeneity by single molecule methods in the revised Introduction. It is thus more specific to the present work of characterizing structural dynamics of HOPS by using single molecule GIET with extended sensitive range.

A few comments should be addressed:– Figure 2 is the core of the paper and the ability to validate the distances of the ssDNA designed as nanorulers sustains all the remaining results. If the results in Figure 2 are questionable, then all other results would be problematic. So, it is important to question why the authors need to include undefined parameters to match their results to the ground truth distance measurements provided by the DNA nanoruler system. The consideration of the tilting parameter is unfortunate and would be important to find a system in which ground truth can be measured without data processing and calculation to match model to experimental data.

We agree with the reviewer that a calibration without a variable parameter would be preferable. However, we have previously confirmed the validity of our electromagnetic simulations of GIET efficiencies using coating with silica layers with different thicknesses (Ghosh et al., 2019). Based on this “ground truth”, we here needed to confirm the possibility of applying this method for distance measurements over a large dynamic range under biologically relevant conditions on a graphene-supported lipid monolayer. This was achieved by using the DNA nanoruler system, for which a robust tilting angle of 43±1 degree was obtained for both intensity and lifetime measurements. In addition, the structural dynamics observed for a 35mer DNA after hybridizing with the 15mer blocker strand confirmed a very similar tilting angle of 45 degree. These self-consistent results furthermore support our calibration by introducing the tilting angle. While additional approaches using cholesterol-modified DNA origami could be envisaged for a more robust distance measurement, establishing such a method is beyond the scope of this study. Rather, we took up the suggestion from reviewer 1 and added one more calibration data using dye-labeled lipid OG488-DHPE (Figure 2—figure supplement 1). When use the additional data point for global fitting, we have obtained the same tilting angle as before, i.e. average 43±1 degree. The new results furthermore confirm the robustness of calibration. In the revised manuscript, we have updated the calibration curve shown in Figure 2B, Figure 2—figure supplement 1, and Supplementary file 1.

– Also is not clear why the authors rely on the ratio between the measurements generated from graphene vs silica layers?

Our measurements are based on the quantum yield (intensity) and the excited state decay kinetics (fluorescence lifetime) as key photophysical properties of the fluorophore that are altered in vicinity of graphene. These properties, however, are also affected by the molecular environment of the fluorophore – as nicely demonstrated by the different *τ_0_* of FAM when attached to 3´and 5´ ends of the oligonucleotides (cf. Supplementary file 1C). Such effects were reliably referenced out by measuring intensities (*I_0_*) and lifetimes (*τ_0_*) under exactly the same conditions (i.e. for the same molecules at a lipid-water interface) but in the absence of GIET. Therefore, we determined *I_0_* and *τ_0_* on silica-supported membranes, thus ensuring that the ratios *I_G_/I_0_* and *τ_G_/τ_0_* only comprise GIET-based quenching effects. Such reference measurements were particularly important for intensity measurements, which strongly depend on molecule densities and acquisition properties that may vary between different measurements. To robustly consider these effects, we applied simultaneous label-free and fluorescence detection side-by-side on silica and graphene surfaces.

In the revised manuscript, we have added an explanation in Materials and methods: Electromagnetic simulation of GIET to clarify this point.

There should be a theoretical explanation for these statements:– "Strikingly, the obtained correlation of IG/I0 vs d matches the predicted GIET curve well with a globally fitted tilting angle of (43 ±1)o (Figure 2B), confirming the distance-dependent GIET on graphene-supported lipid monolayer."– "Ratios of fluorescence lifetimes on graphene (τG) to those on glass (τ0) closely matched the corresponding intensity ratios, which also confirmed the simulated distance dependent GIET efficiency (Figure 2B, Supplementary file 1C)".

These two comments are related to whether the ratios of fluorescence intensity and lifetime can be used for GIET experiments. As mentioned above, the ratio of intensities measured on graphene and glass normalizes the influence of experimental conditions, and the fluorescence intensity is proportional to the lifetime since there is no static quenching in our experiments. Thus the relative fluorescence intensity and relative lifetime both allow reliable quantification of the distance.

We have now added the necessary explanation in the revised Materials and methods: Electromagnetic simulation of GIET.

– Similarly, for longer distances >10nm it would be important to have a DNA ruler than could directly be measured as ground truth using GIET on graphene-supported lipid monolayer without needing comparison with silica substrate measurements.

As we have answered to the previous comments, measurements on silica were performed to robustly determine *I_0_* and *τ_0_*, which were applied for all rulers, independent on the fluorophore distance from the surface. Such reference measurements are integral part of determining the GIET efficiencies and thus the distances from the graphene surface.

In the revised manuscript, we have updated the results of calibration shown in Figure 2B, Figure 2—figure supplement 1 and Supplementary file 1. We have also added the necessary explanations in Materials and methods: Electromagnetic simulation of GIET for using the ratios.

– Also it is interesting that a similar angle of 45o is found for calibration curve as well as for the measurements of 35mer DNA nanoruler: "Comparing to the length of the 35mer DNA nanoruler (11.9 nm), the height corresponds to a tilting angle α = 45{degree sign}, which is in good agreement with the tilting angle α obtained for the calibration." Is there an explanation for this finding/coincidence?

The “coincidence” as reviewer pointed out is indeed a very important evidence supporting the validation by using the DNA nanorulers. We have shown global fitting of 7 intensity measurements (now 8 with the additional data at position zero) and 7 lifetime measurements to a highly conserved 43±1 degree tilting angle. Based on the calibration, an independent experiment for exploring the structural dynamics of 35mer DNA obtains a tilting angle of 45 degree. The self-consistent result of the tilting angles, i.e. 45 degree vs the globally fitted 43±1 degree, thus confirm the justification of introducing tilting of DNA nanorulers in the calibration.

In the revised manuscript, we have now emphasized this point.

Reviewer #3:In brief, this work highlights GIET as a unique spectroscopic ruler to reveal the axial orientation and dynamics of macromolecular complexes at biological membranes with sub nanometer resolution, which is a very significant contribution to the mechanistic understanding of numerous key biological processes.Surface plasmon enhancement of electromagnetic field on metals, semiconductors and graphene has been implemented in various biosensing applications, covering a spectral region from UV-visible to IR. Comparing to the broad applications of surface enhancement, the nearfield radiationless electromagnetic coupling between the surface plasmon and excited fluorephores is less harnessed for biosensing. The situation is changing. Metal induced energy transfer (MIET) and graphene-based MIET (equivalent to “GIET” of graphene induced energy transfer) have emerged recently. In this manuscript, Fullbrunn et al. have set up the first step of using GIET in the field of protein structural biology. In contrast of detecting the structures of DNA origami or likely robust antibody-based biomolecular analysis, the current work draws a high standard by using lipidated membrane association protein and a tricky multiprotein complex. The authors' strategy is straightforward which achieved their goals efficiently: biofunctionalizing graphene on glass, validating the distance dependent GIET, and determining the structure of protein complexes. With the interdisciplinary expertise, the authors successfully used GIET as a spectroscopic ruler mapping the axial distance of the ~20 nm long HOPS complex on graphene. Very impressively, the structural dynamics of HOPS complexes was quantified by single molecule GIET on graphene-coated microscopy coverslips. I found this research was very well designed and carried out carefully. The manuscript was written in a good logical flow. The conclusions were supported by sufficient figures, tables and their supplementary.The longer sensitive range of GIET than FRET is indeed very suitable to obtain the structural overview of many extended proteins. As demonstrated in this work, GIET could be a powerful tool for determining the structure of membrane proteins. However, being a surface plasmon-based method, GIET is limited by immobilizing target proteins on the surface. From this aspect, one must point out GIET is not as feasible as FRET for applications in solution. Another disadvantage of the current method is that the coating on graphene is a non-fluidic lipid monolayer (See FRAP experiments in subsection “Axial reorganization at the lipid monolayer surface can be quantified by GIET” and Figure 2—figure supplement 5, Figure 3—figure supplement 4). The authors has noticed in the conclusions “GIET will allow to dissect.… large protein complexes on membranes upon interaction with membrane-bound ligands, and thus test molecular hypotheses in real time”. Actually, instead of considering “interaction of membrane-bound ligands”, surface architectures of fluidic lipid bilayer will be urgently needed for the applications of a large number of biologically important transmembrane proteins.

It is true that GIET is a surface-based structural analysis tool and therefore we clearly identified its potential for structural analysis of protein complexes in the context of membranes. We also agree with the importance of extending this methodology towards fluid lipid bilayer architectures stressed by the reviewer and we are currently pushing our efforts into this direction. For our proof-of-concept experiments, however, the well-defined monolayer architecture used in this study clearly provided important advantages for quantifying distances with minimum bias.